# TAH-QUANT: EFFECTIVE ACTIVATION QUANTIZATION IN PIPELINE PARALLELISM OVER SLOW NETWORK

## ABSTRACT

Decentralized training of large language models offers the opportunity to pool computational resources across geographically distributed participants but faces significant network communication bottlenecks, particularly in pipeline-parallel settings. While pipeline parallelism partitions model layers across devices to handle large-scale models, it necessitates frequent communication of intermediate activations, creating challenges when network bandwidth is limited. Existing activation compression methods, such as AQ-SGD, mitigate quantization-induced errors through error compensation but impose prohibitive memory overhead by requiring storage of previous activations. To address these issues, we introduce TAH-QUANT (**T**ile-wise **A**daptive **H**adamard **Quant**ization), a novel activation quantization framework designed specifically for pipeline parallelism. Our approach integrates fine-grained tile-wise quantization for precise control, entropy-guided token-level adaptive bit allocation for optimal bit usage, and a Hadamard-based transform with pivot element swapping to effectively suppress quantization outliers. We further provide a theoretical analysis, proving that pipeline parallel training equipped with TAH-QUANT maintains a convergence rate of $\mathcal{O}(1/\sqrt{T})$, matching that of vanilla stochastic gradient descent. Extensive experiments on diverse LLM tasks demonstrate that TAH-QUANT achieves aggressive activation quantization (3-4 bits) ratio, which provides up to $1.33\times$ end-to-end speedup without compromising training convergence, matches state-of-the-art methods, incurs no extra memory overhead, and generalizes well across different training scenarios.

## 1 INTRODUCTION

Decentralized or open collaborative training of large language models (LLMs) has recently gained significant attention as it enables pooling computational resources across multiple geo-distributed participants, thus facilitating training of models that exceed the capacity of any single resource contributor Ryabinin & Gusev (2020); Yuan et al. (2022); Gandhi et al. (2024). However, a major barrier to these approaches is network communication: unlike specialized clusters equipped with high-speed interconnects, decentralized settings typically rely on slower networks, severely constraining training efficiency Wang et al. (2022; 2023b). On the other hand, scaling LLM training in state-of-the-art scale necessitates distributed model parallel training — particularly pipeline parallelism Huang et al. (2019); Narayanan et al. (2019; 2021), which partitions model layers across multiple stages to support training LLMs with billions of parameters. Yet, pipeline parallelism inherently requires frequent transmission of activations and their corresponding gradients between adjacent pipeline stages. In this paper, we explore *how to effectively compress the communication volume to accommodate pipeline parallelism over slow network links.*

Enabling efficient activation compression for pipeline parallelism over slow network connections has significant implications for democratizing large-scale LLM training Yuan et al. (2022); Wang et al. (2022). Currently, the capability to train state-of-the-art models remains concentrated among institutions equipped with specialized high-performance computing resources. Effectively addressing network communication bottlenecks would substantially reduce barriers to participation, allowing a broader array of contributors, including universities, startups, and individuals, to collaboratively train or fine-tune LLMs Douillard et al. (2025).

On the other hand, a significant obstacle arises from the fact that naive activation compression (e.g., quantization Han et al. (2016); Hubara et al. (2017)) methods can negatively affect training conver-

gence. Unlike gradient compression in data parallelism — where quantization errors typically behave as unbiased noise compatible with optimization procedures, compressing intermediate activations directly influences the neural network's forward computation, consequently introducing bias into gradient estimates in the later backward propagation Evans & Aamodt (2021); Chakrabarti & Moseley (2019). Specifically, in pipeline parallel training, compression errors incurred during activation transmission propagate through nonlinear transformations and can distort gradient calculations during the backward pass Wang et al. (2022). Thus, aggressively reducing activation precision without careful management will result in performance degradation or even training divergence.

To restrict the error propagation introduced by activation quantization, prior efforts, such as AQ-SGD Wang et al. (2022), have attempted to address this issue by compressing the changes in activations between training epochs rather than the activations themselves, thereby providing theoretical convergence guarantees leveraging the help from the error compensation. Although effective in preserving model accuracy, AQ-SGD requires storing previous activations for the whole dataset to compute these changes, resulting in substantial memory overhead. Such an approach poses practical limitations, especially in resource-constrained environments for the large-volumes of training data where storage capacity and system complexity are critical considerations.

In this paper, we solve this problem with a new approach for effective activation quantization in pipeline parallelism. In particular, we make the following key contributions:

**Contribution 1.** We propose TAH-QUANT (**T**ile-wise **A**daptive **H**adamard **Quant**ization), an activation quantization approach to alleviate communication bottlenecks in pipeline-parallel training of LLMs. Specifically, our method includes: (**i**) a fine-grained, tile-wise group quantization technique for localized precision control, effectively limiting quantization error; (**ii**) an entropy-guided, token-level adaptive bit allocation method that dynamically assigns precision based on activation distribution characteristics, further optimizing the compression efficiency; and (**iii**) a Hadamard-based outlier suppression transform enhanced by a pivot element swap, which effectively mitigates quantization errors arising from extreme activation values. Collectively, these carefully designed techniques enable efficient, accurate low-bit quantization of activations, substantially improving the practicality of decentralized and collaborative LLM training.

**Contribution 2.** Given the advances of TAH-QUANT, we further conduct some case studies about the quantization error in pipeline parallel training. Then we conduct some theoretical analysis under standard stochastic optimization assumptions, along with an additional assumption that characterizes the behavior of TAH-QUANT's quantization error, which is empirically validated by our extensive experimental results. Concretely, we theoretically prove that pipeline parallel training equipped with TAH-QUANT converges at a rate of $\mathcal{O}(1/\sqrt{T})$, matching the convergence rate of vanilla SGD.

**Contribution 3.** We then conduct extensive experiments on various LLM training tasks (e.g., including `GPT2-XL`, `Qwen2.5-3B`). We show that TAH-QUANT can aggressively quantize activations to 3-4 bits without sacrificing convergence performance similar to the state-of-the-art, *i.e.*, AQ-SGD, without introducing any storage overheads, and is generally applicable to different training tasks.

## 2 PRELIMINARY AND RELATED WORK

**Decentralized training of LLM.** Decentralized training of LLMs has garnered significant attention as an interesting attempt to democratize access to large-scale LLM training development Ryabinin & Gusev (2020); Borzunov et al. (2022; 2023); Gandhi et al. (2024); Blagoev et al. (2025). Early efforts demonstrated the feasibility of collaborative training across geographically distributed participants with constrained resources under the scope of data parallelism Diskin et al. (2021); Borzunov et al. (2022), where various effective gradient compression methods have been explored Wang et al. (2023b). To further scale out the training computation, more advanced moded parallel stragies have been integrated Yuan et al. (2022); Ryabinin et al. (2023); Lu et al. (2024); Strati et al. (2024), for example, Yuan et al. Yuan et al. (2022) addressed the challenges of training foundation models in heterogeneous environments by introducing a scheduling algorithm that optimally allocates computational tasks across decentralized GPUs; Ryabinin et al. Ryabinin et al. (2023) proposed SWARM Parallelism, where temporary randomized pipelines between nodes are adaptively rebalanced to handle dynamic efficient training of large Transformer models using preemptive instances with limited network bandwidth. Douillard et al. Douillard et al. (2025) introduces enhancements to the DiLoCo framework

by employing sequential parameter synchronization, overlapping communication with computation, and quantized data exchange.

**Activation compression in training.** Activation compression techniques have been studied to reduce memory and computational overhead in neural network training Liu et al. (2021); Bersatti et al. (2020); Georgiadis (2019); Fu et al. (2020); Liu et al. (2022); Chen et al.; Bian et al. (2024). Concretely, inherent sparsity in activation has been studied to minimize storage and computation in neural networking. For example, Zhang et al. Zhang et al. (2024) investigate the natural occurrence of sparse activations in pre-trained Transformers and dynamically alternate between sparse and dense training phases to enhance pre-training efficiency Rhu et al. (2018); Jiang et al. (2022); Zhang et al. (2024); Li et al.. On the other hand, quantization-based methods Evans et al. (2020); Liu et al. (2021); Wang et al. (2023a) reduce the precision of activations to lower bit-widths, thereby decreasing memory usage. For example, Han et al. Han et al. (2016) presented Deep Compression, combining pruning, trained quantization, and Huffman coding. Hubara et al. Hubara et al. (2017) explore training neural networks with low-precision weights and activations. Chakrabarti et al. Chakrabarti & Moseley (2019) propose backpropagation with approximate activations for memory-efficient training. Chen et al. Chen et al. (2021a) introduced ActNN, employing 2-bit activation compressed training.

**Quantization for LLM.** Quantization has emerged as a key technique for serving LLMs efficiently by reducing the precision of weights Lin et al. (2024b); Frantar et al. (2022), activations Xiao et al. (2023), and KV-cache Liu et al. (2024b) for the process of generative inference, parameter-efficient fine-tuning Dettmers et al. (2023), and large-scale pretraining You et al. (2024); Liu et al. (2024a). For example, AWQ Lin et al. (2024b) quantizes the LLM weights by identifying a small subset of "salient" weight channels and scales them up before quantization, thereby preserving accuracy even at 4-bit weight precision. KIVI Liu et al. (2024b) proposes a tuning-free 2-bit quantization of the KV cache (with per-channel asymmetric scaling), dramatically reducing memory and enabling longer context lengths with negligible impact on generation quality. QLoRA Dettmers et al. (2023) demonstrated that a 4-bit quantized base model can be *fine-tuned* via low-rank adapters to reach the same performance as full `FP16` fine-tuning. LLM-QAT Liu et al. (2024a) introduces a data-free QAT scheme that allows 4-bit quantization of weights, activations, and even the KV cache while preserving performance for training. One essential problem in such quantization methods is how to effectively resolve the issues of outliers in the quantization group Lin et al. (2024a); Hu et al. (2025); You et al. (2024). An especially simple yet effective transformation for outlier suppression is the *Hadamard transform* Theodoridis & Koutroumbas (2009). Formally, the $N \times N$ Hadamard matrix $\mathbf{H}_N \in \pm1^{N \times N}$ is defined such that $\mathbf{H}_N \mathbf{H}_N^T = N \mathbf{I}_N$ (so $\frac{1}{\sqrt{n}} \mathbf{H}_N$ is an orthonormal matrix). Multiplying a vector with dimension $N$ by $\mathbf{H}_N$ will evenly redistribute the vector's components across $N$ dimensions.

## 3 ACTIVATION QUANTIZATION

In pipeline-parallel training Huang et al. (2019); Narayanan et al. (2019; 2021) of LLM, intermediate activations must be communicated between devices. This communication can become a significant bottleneck on slow interconnects. To alleviate this overhead by quantization-based compression, we leverage three main carefully-designed mechanisms *in order to reduce the quantization error*, including: (**i**) fine-grained tile-wise group quantization for localized precision control (Section 3.1); (**ii**) an entropy-guided token-level adaptive bit-width allocation (Section 3.2); and (**iii**) a Hadamard-transform-based outlier suppression with a pivot element swap (Section 3.3). We also discuss how we integrate the proposed TAH-QUANT quantization method in pipeline parallel training in Section 3.4. We enumerate the details below.

### 3.1 FINE-GRAINED TILE-WISE GROUP QUANTIZATION

First, we introduce a fine-grained, tile-wise group quantization scheme for localized precision control. Specifically, instead of quantizing the entire activation tensor with a single set of parameters, we partition it into small tiles and quantize each tile independently. For example, consider an activation tensor $\mathbf{a}$ of shape $B \times S \times C$, i.e., $\mathbf{a} \in \mathbb{R}^{B \times S \times C}$, where $B$, $S$, and $C$ denote the batch size, sequence length, and number of channels (i.e., model dimension), respectively. We partition this tensor along the channel dimension into multiple tiles by grouping contiguous channels within each token. Each such tile (i.e., quantization group) can be noted as $\mathbf{a}_{i,j,t} \in \mathbb{R}^G$, where $G$ is the quantization group size determined by $G = \frac{C}{N_t}$, $N_t$ is the number of partitions of all the channels, and $i = 1, ..., B$,

$j = 1, ..., S, t = 1, ..., N_t$ are the indices for each tile-wise quantization group. Note that each tile will form a separate quantization group with its own scale and zero-point. This fine-grained approach ensures that each group is quantized using an optimal dynamic range, greatly improving accuracy in low-bit settings. By confining quantization error to these small groups, we preserve more information compared to coarse, whole-tensor quantization, which can be dominated by a few extreme values.

## 3.2 Token-Level Adaptive Bit Allocation

The fine-grained grouping addresses local range variation; however, the value associated with different tokens in the activation tensor may still have varying importance. Toward this end, we introduce an entropy-based, token-level quantized precision allocation strategy that dynamically adjusts the quantization bit width for the activation values associated with each token's groups.

Concretely, for each token's activation vector $\mathbf{a}_{i,j} \in \mathbb{R}^C$, we compute the entropy $\mathcal{H}(\mathbf{a}_{i,j})$ of its normalized magnitude distribution to quantify how the activation's energy is spread across channels. To obtain this distribution, we take the absolute value of each channel and normalize:

$$p_k = \frac{|a_{i,j,k}|}{\|\mathbf{a}_{i,j}\|_1 + \epsilon} \quad k = 1, ..., C \tag{1}$$

Where $\|\mathbf{a}_{i,j}\|_1$ is the L1 norm of the activation vector and $\epsilon$ is a small positive constant for numerical stability. We formulate the entropy of this distribution associated with the activation vector $\mathbf{a}_{i,j}$ as

$$\mathcal{H}(\mathbf{a}_{i,j}) = \sum_{k=1}^{C} p_k \log(p_k + \varsigma) \tag{2}$$

Where $\varsigma$ is another small positive constant to avoid zero values. Note that the entropy $\mathcal{H}(\mathbf{a}_{i,j})$ will be high when the magnitudes are evenly spread across all channels (i.e. $\mathbf{a}_{i,j}$ has an approximately uniform distribution with no dominant feature) and low when the activations are concentrated or structured (e.g., dominated by a few channels or containing an outlier). We leverage this entropy measurement to guide the bit-width allocation for quantization. Intuitively, if all the channels of a token are similarly scaled (high entropy), compressing it too aggressively could compromise the detailed information of the features at once. We therefore assign these high-entropy tokens a higher bit width (i.e., 4 bits, `INT4`) to preserve precision. Conversely, if a token's activation is dominated by a few large components (low entropy), it contains a strong outlier structure that our transform (as we will introduce in Section 3.3) can effectively resolve. Thus, we can quantize more aggressively (i.e., 3 bits, `INT3`) without incurring significant loss. In practice, we determine each token's bit allocation by ranking $\mathcal{H}(\mathbf{a}_{i,j})$, where top-$p\%$ of the tokens are quantized to `INT4` while the rest are quantized to `INT3`. Note that all of the tiles belonging to the same token share the same bit allocation results. By tailoring the precision to each token's content, we maximize overall compression efficiency under a fixed bit budget.

## 3.3 Hadamard-Based Outlier Suppression Transform

Outliers in the activation values can severely degrade the accuracy of low-bit quantization even within a small quantization group. To mitigate quantization error caused by extreme outliers in activation groups, we propose an adaptive Hadamard transform strategy, which consists of three steps: (**i**) a *heuristic-based outlier detection* to decide if transform is needed, (**ii**) a *Hadamard transform with pivot element swap* to redistribute the outlier values in the quantization group, and (**iii**) an *asymmetric uniform quantization* of the values in the quantization group.

**Outlier detection heuristic**: Given any quantization group $\mathbf{a}_{i,j,t} = [a_1^{i,j,t}, a_2^{i,j,t}, \ldots, a_G^{i,j,t}] \in \mathbb{R}^G$, where $G = \frac{C}{N_t}$ is the quantization group size. For the rest parts in Section 3.3, we simplify the notation as $\mathbf{a}_{i,j,t} = \boldsymbol{\alpha} = [\alpha_1, \alpha_2, \ldots, \alpha_G] \in \mathbb{R}^G$ to introduce the quantization method within each quantization group. In order to detect whether an outlier is present, we define the following heuristic:

$$r = \frac{|\alpha^{(1)}|}{|\alpha^{(2)}| + \varrho} \tag{3}$$

Where $\alpha^{(1)}$ and $\alpha^{(2)}$ represent the elements in $\boldsymbol{\alpha}$ with the largest and the second largest absolute values, $\varrho$ is a small positive constant. If $r$ exceeds a threshold $\tau$ (empirically, we set $\tau = 2.0$), we will deem $\mathbf{a}_{i,j,t}$ to contain an outlier and apply the Hadamard-based transform as we will introduce below; otherwise, we skip this transform for that tile.

---

**Algorithm 1** TAH-QUANT in a two-stage pipeline parallel training.

---

1: **Initialize:** sub-network $a(-)$ weights $\mathbf{x}^{(a)}$, sub-network $b(-)$ weights $\mathbf{x}^{(b)}$, optimizer $\rho$.
2: **for** t = 1, . . . , T **do**
3:     Randomly sample training batch $\xi_t$.
      // Forward propagation:
4:     Machine $a$ sends the quantized output activations $Q_{\text{TAH-QUANT}}\left(a(\xi_t, \mathbf{x}_t^{(a)})\right)$ to Machine $b$.
5:     Machine $b$ dequantizes the received activation $Q_{\text{TAH-QUANT}}\left(a(\xi_t, \mathbf{x}_t^{(a)})\right)$.
      // Backward propagation:
6:     Machine $b$ sends the quantized gradients w.r.t the activations $Q_{\text{NAIVE}}\left(\nabla_a(F \circ b)|_{\xi_t}\right)$ back to Machine $a$.
7:     Machine $a$ dequantizes the received gradient w.r.t the activations $Q_{\text{NAIVE}}\left(\nabla_a(F \circ b)|_{\xi_t}\right)$.
      // Parameter updates:
8:     Machine $a$ update its parameter by gradients $\hat{\mathbf{g}}^t\left(\mathbf{x}^{(a)}\right)$ using optimizer $\rho$.
9:     Machine $b$ update its parameter by gradients $\hat{\mathbf{g}}^t\left(\mathbf{x}^{(b)}\right)$ using optimizer $\rho$.
10: **end for**
11: **Output:** $\mathbf{x} = (\mathbf{x}_T^{(a)}, \mathbf{x}_T^{(b)})$

---

**Hadamard transform with pivot element swap**: For a group identified to have an outlier, we perform a *pivot element swap* to align the pivot value (the element with the largest absolute value) with the Hadamard matrix structure. Let $d = \arg\max_k |\alpha_k|$ be the index of the pivot element (i.e., $\alpha_d = \alpha^{(1)}$). We construct a permutation matrix $\mathbf{P}_d \in \mathbb{R}^{G \times G}$ that swaps the first and $d$-th coordinates, which yields a permuted vector by multiplying this permutation matrix:

$$[\alpha_d, \alpha_2, \ldots, \alpha_1, \ldots, \alpha_G] = [\alpha_1, \alpha_2, \ldots, \alpha_G]\mathbf{P}_d = \boldsymbol{\alpha}\mathbf{P}_d$$

Next, we multiply this transformed vector by a Hadamard matrix $\mathbf{H}_G \in \pm 1^{G \times G}$ to redistribute the values and resolve the issue of outliers:

$$\dot{\boldsymbol{\alpha}} = \boldsymbol{\alpha}\mathbf{P}_d \frac{1}{\sqrt{G}}\mathbf{H}_G \tag{4}$$

After applying this transform, the extreme value in the original $\boldsymbol{\alpha}$ will be redistributed across all components in the transformed vector $\dot{\boldsymbol{\alpha}}$. This transform greatly reduces the dynamic range of the group: the formerly pivot value is no longer isolated in a single position, yielding a more balanced tile for the activation vector. As a result, the quantization error can be reduced, since a tighter quantization scale can represent the values with higher precision. Notably, because $\mathbf{H}_G$ is orthogonal (i.e., $\mathbf{H}_G \mathbf{H}_G^T = G\mathbf{I}_G$), we can later invert the transform by applying $\mathbf{H}_G^T$ to the de-quantized values when recovery of the original domain is required.

**Asymmetric uniform quantization**: After the above two steps, the activation values should be uniformly distributed and centered if the outlier issue once existed. Thus, we can apply the a standard asymmetric quantizer (i.e., You et al. (2024)) — if the computed heuristic $r \leq \tau$, we apply this quantizer for the original vector $\boldsymbol{\alpha}$; otherwise, we apply this quantizer for the transformed vector $\dot{\boldsymbol{\alpha}}$.

### 3.4 TAH-QUANT IN PIPELINE PARALLEL TRAINING

Given the carefully designed TAH-QUANT quantization method, it is straightforward to integrate it into the standard pipeline parallel training. We illustrate this process in Algorithm 1. For clarity, we present it using a two-stage pipeline, which can be easily extended to an arbitrary number of stages. Note that after some empirical verification, we follow the design of `AQ-SGD` —- we apply the naive quantization during the backward pass, where more bits can be allocated since more computation load in backward propagation provides more slots during the system optimization of computation-communication overlapping Yuan et al. (2022).

## 4 THEORETICAL ANALYSIS

In this section, we present convergence guarantees for the proposed TAH-QUANT algorithm, which aims to solve the following stochastic optimization problem in a pipeline-parallel fashion:

$$\min_{\mathbf{x} \in \mathbb{R}^d} \quad \mathbb{E}_{\xi \in \mathcal{D}}[F(\mathbf{x}; \xi)] \tag{5}$$

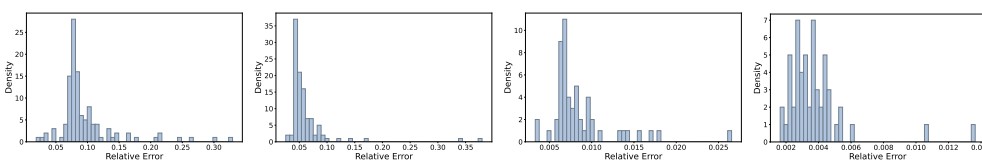

(a) step wise; tile size 64 (b) step wise; tile size 32 (c) full dataset; tile size 64 (d) full dataset; tile size
;80% INT4 20% INT3.    ;80% INT4 20% INT3.    ;80% INT4 20% INT3.    32 ;80% INT4 20% INT3.

Figure 1: Empirical justification of Assumption 4

where $\mathbf{x}$ denotes the model weights distributed across different pipelines, and $\xi$ represents random data drawn from the distribution $\mathcal{D}$. We denote $\nabla F(\mathbf{x}; \xi)$ as the stochastic gradient and $\nabla f(\mathbf{x})$ as the full-batch gradient. Without loss of generality, we consider using momentum SGD as the optimizer $\rho$ in Algorithm 1:

$$\mathbf{m}^t = (1 - \beta_1)\mathbf{m}^{t-1} + \beta_1\hat{\mathbf{g}}^t, \qquad \mathbf{x}^{t+1} = \mathbf{x}^t - \eta\mathbf{m}^t, \tag{6}$$

where $\beta_1 \in (0, 1)$ is the momentum coefficient and $\eta$ is the learning rate. The vector $\hat{\mathbf{g}}^t$ is a quantized estimate of the stochastic gradient $\nabla F(\mathbf{x}^t; \xi^t)$, obtained through Lines 3–7 of Algorithm 1. Specifically, it takes the form $\left(\hat{\mathbf{g}}^t(\mathbf{x}^{(a_1)}), \hat{\mathbf{g}}^t(\mathbf{x}^{(a_2)}), \ldots, \hat{\mathbf{g}}^t(\mathbf{x}^{(a_N)})\right)$, where $a_1, a_2, \ldots, a_N$ index the machines in the pipeline-parallel system. Our analysis can be extended to Adam optimizer with a few more involved derivations.

### 4.1 ASSUMPTIONS

**Assumption 1** (Lower Boundedness). *The loss function* $f : \mathbb{R}^d \to \mathbb{R}$ *satisfies* $\inf_{\mathbf{x}\in\mathbb{R}^d} f(\mathbf{x}) > -\infty$.

**Assumption 2** (L-Smoothness). *The loss function* $f$ *is L-smooth, i.e., it holds for any* $\mathbf{x}, \mathbf{y} \in \mathbb{R}^d$ *that*

$$\|\nabla f(\mathbf{x}) - \nabla f(\mathbf{y})\|_2 \leq L\|\mathbf{x} - \mathbf{y}\|_2.$$

**Assumption 3** (Stochastic Gradient). *We assume the stochastic gradient oracle satisfies*

$$\mathbb{E}[\nabla F(\mathbf{x}^t; \xi^t)] = \nabla f(\mathbf{x}^t), \qquad \mathbb{E}[\|\nabla F(\mathbf{x}^t; \xi^t) - \nabla f(\mathbf{x}^t)\|^2] \leq \sigma^2, \tag{7}$$

*for some* $\sigma > 0$.

Assumptions 1-3 are standard assumptions commonly used in stochastic optimization. The following assumption states that gradient quantization through TAH-QUANT proposed in Algorithm 1 does not introduce significant distortion to the true stochastic gradient.

**Assumption 4** (Quantization Error). *Let* $\mathbf{g}^t$ *denote the original stochastic gradient* $\nabla F(\mathbf{x}^t, \xi^t)$, *and* $\hat{\mathbf{g}}^t$ *denote the quantized stochastic gradient obtained through* TAH-QUANT. *It holds that*

$$\|\hat{\mathbf{g}}^t - \mathbf{g}^t\|^2 \leq (1 - \delta)\|\mathbf{g}^t\|^2, \tag{8}$$

$$\|\mathbb{E}_{\xi^t\sim\mathcal{D}}[\hat{\mathbf{g}}^t] - \nabla f(\mathbf{x}^t)\|^2 \leq (1 - \delta)\|\nabla f(\mathbf{x}^t)\|^2, \tag{9}$$

*for some* $\delta \in (0, 1]$.

The above assumption ensures that the quantized gradient $\hat{\mathbf{g}}$ remains close to the true gradient $\mathbf{g}$, with their closeness measured by the quantization coefficient $\delta$. A larger $\delta$ (*i.e.*, $\delta \to 1$) indicates a smaller quantization error. When $\delta = 1$, we have $\hat{\mathbf{g}} = \mathbf{g}$, implying no quantization error.

**Empirical justification of Assumption 4.** We now empirically verify that TAH-QUANT satisfies Assumption 4. To validate inequality (8), we conduct fine-tuning experiments on the Gemma2-2B model using the Math-7K dataset. At each training step, we compute the relative error $\|\hat{\mathbf{g}}^t - \mathbf{g}^t\|^2/\|\mathbf{g}^t\|^2$, as shown in Figure 1a1b. The results indicate that the relative errors remain below 0.4 across all steps, confirming the validity of (8) with $\delta = 0.6$. To validate inequality (9), we conduct experiments on the same model and dataset. At each step, we compute both the expected compressed gradient $\mathbb{E}_{\xi^t\sim\mathcal{D}}[\hat{\mathbf{g}}^t]$ and the full-batch gradient $\nabla f(\mathbf{x}^t)$, and then evaluate the relative error $\|\mathbb{E}_{\xi^t\sim\mathcal{D}}[\hat{\mathbf{g}}^t] - \nabla f(\mathbf{x}^t)\|^2/\|\nabla f(\mathbf{x}^t)\|^2$, as shown in Figure 1c1d. All relative errors are below 0.1, confirming the validity of (9) with $\delta = 0.9$. In both experiments, we use tile sizes of 64 and 32, with 80% INT4 and 20% INT3 quantization. These experiments demonstrate the effectiveness of TAH-QUANT, which quantizes variables to smaller sizes without incurring significant errors.

### 4.2 CONVERGENCE GUARANTEES

Under the above assumptions, we are ready to provide convergence guarantees of our proposed TAH-QUANT method.

**Theorem 4.1.** *Under Assumptions 1 - 4, if* $\beta_1 \in \left(0, \frac{\delta}{24-12\delta}\right)$, $\delta \in (0,1)$ *and* $\eta \leq \min\left\{\frac{1}{2L}, \frac{\beta_1}{L} \cdot \sqrt{\frac{\delta}{8}}\right\}$, TAH-QUANT *with momentum SGD converges as*

$$\frac{1}{T+1}\sum_{t=0}^{T}\mathbb{E}[\|\nabla f(\mathbf{x}^t)\|_2^2] \leq \frac{8[f(\mathbf{x}^0) - \inf_{\mathbf{x}} f(\mathbf{x})]}{\delta\eta(T+1)} + \frac{8\|\mathbf{m}^0 - \nabla f(\mathbf{x}^0)\|_2^2}{\delta\beta_1(T+1)} + \frac{24\beta_1\sigma^2}{\delta}.$$

**Corollary 4.1.** *Under Assumptions 1-4, if we choose* $\beta_1 = \left(\frac{24}{\delta} + \sigma\sqrt{\frac{\delta^{1/2}(T+1)}{L\Delta}}\right)^{-1}$, $\eta = \left(2L + \frac{2^{3/2}L}{\delta^{1/2}\beta_1}\right)^{-1}$, TAH-QUANT *with momentum SGD converges as (Proofs are in Appendix D)*

$$\frac{1}{T+1}\sum_{t=0}^{T}\mathbb{E}[\|\nabla f(\mathbf{x}^t)\|_2^2] = \mathcal{O}\left(\frac{L\Delta}{\delta^{5/2}(T+1)} + \sqrt{\frac{L\Delta\sigma^2}{\delta^{5/2}(T+1)}}\right),$$

*where* $\Delta := f(\mathbf{x}^0) - \inf_{\mathbf{x}} f(\mathbf{x}) + (\delta/L) \cdot \|\mathbf{m}^0 - \nabla f(\mathbf{x}^0)\|_2^2$.

**Remark.** Corollary 4.1 yields three key implications. First, it guarantees that the proposed TAH-QUANT algorithm converges to a stationary solution of problem (5). Second, it shows that TAH-QUANT achieves a convergence rate of $\mathcal{O}(1/\sqrt{T})$, matching that of vanilla momentum SGD without gradient quantization. This demonstrates that TAH-QUANT effectively preserves the valuable gradient information during quantization. Third, the theorem indicates that the convergence rate is affected by the quantization error, quantified by the coefficient $\delta$. This is consistent with our expectations. Since TAH-QUANT maintains a relatively large $\delta$ (i.e., close to 1), the quantization error remains moderate and does not significantly slow convergence.

## 5 EVALUATION

We demonstrate that TAH-QUANT significantly accelerates LLM training over slow network connections. Specifically, we show that: (i) on seven representative benchmark tasks, TAH-QUANT enables aggressive quantization of activations and backward gradients without compromising convergence performance or incurring notable additional system overhead (Section 5.2); and (ii) the effectiveness of our system design is validated through a series of carefully designed ablation studies (Section 5.3).

### 5.1 EXPERIMENTAL SETUP

**Datasets and benchmarks**. We evaluate the proposed method on five distinct training scenarios spanning language modeling (on both general and domain-specific text) and instruction-following tasks. Specifically, we fine-tune `GPT-2XL` (1.5B parameters) on (i) `WikiText-2`, a standard Wikipedia-based language modeling benchmark, and (ii) `ArXiv21`, a corpus of research paper abstracts from arXiv. To assess performance on instruction data, we fine-tune `Qwen2.5-3B` (3B parameters) on (iii) `Magicoder-Evol-Instruct-110K`, a dataset of 110k code-related instruction-response pairs, and (iv) `Open-Platyups`, a composite open-source instruction tuning dataset covering multiple domains. Finally, we launch the pretraining of `Qwen2.5-3B` on (v) `C4` common crawl corpus for $6,000$ iterations. These setups cover both general and specialized tasks, as well as supervised instruction tuning and LLM pretaining.

**Distributed cluster**. All experiments are conducted on UCloud ucl using a distributed cluster of 8 instances, each equipped with an Nvidia RTX 3090 GPU. Each model is partitioned into 8 pipeline stages (one stage per GPU) to execute pipeline parallelism. The cluster's default interconnect bandwidth is 10 Gbps. To emulate slow-network conditions, we throttle inter-instance communication using Linux traffic control (tc), artificially limiting the bandwidth during training.

**Baselines**. We compare our approach with two baseline communication strategies[1].

- `FP32/FP16`, which uses full-precision 32-bit floating point (in Tasks (i) and (ii)) or 16-bit floating point (in Tasks (iii), (iv), and (v)) communication with no compression.

---

[1] Each baseline is integrated into the same pipeline parallel training setup for fair comparison.

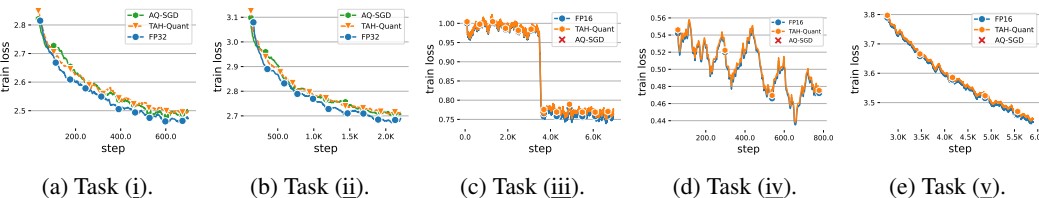

(a) Task (i).     (b) Task (ii).     (c) Task (iii).     (d) Task (iv).     (e) Task (v).

Figure 2: The training convergence for each task (loss vs. steps). Task (i): `GPT-2XL` on `WikiText-2`; Task (ii): `GPT-2XL` on `ArXiv21`; Task (iii): `Qwen2.5-3B` on `Magicoder-Evol-Instruct-110K`; Task (iv): `Qwen2.5-3B` on `Open-Platyups`; Task (v): `Qwen2.5-3B` on the `C4`.

- `AQ-SGD`, the error-compensated low-bit activation quantization method with theoretical convergence guarantees Wang et al. (2022).

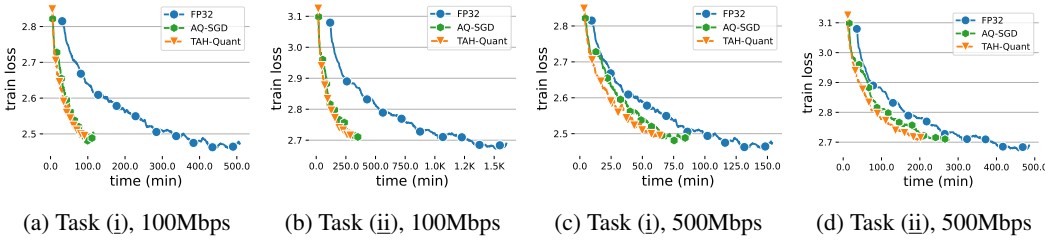

(a) Task (i), 100Mbps    (b) Task (ii), 100Mbps    (c) Task (i), 500Mbps    (d) Task (ii), 500Mbps

Figure 3: End-to-end training performance over different networks

## 5.2 END-TO-END PERFORMANCE RESULTS

To systematically evaluate the performance of the proposed TAH-QUANT quantization method, we conduct the experiment and report the corresponding results in terms of training convergence and end-to-end training time.

**Convergence.** Figure 2 illustrates the convergence comparisons across tasks, which clearly demonstrates the efficacy and robustness of TAH-QUANT. Specifically, on tasks (i) and (ii), where AQ-SGD is executable due to manageable dataset sizes and multi-epoch training paradigm, TAH-QUANT achieves comparable or slightly superior convergence performance compared to AQ-SGD, highlighting its effective quantization without loss in training quality. Importantly, on larger-scale tasks (i.e., tasks (iii), (iv),

Table 1: Training throughput (tokens per second)) of `GPT2-xl` varying bandwidths.

| Network Bandwidth | `FP32` | `AQ-SGD` fw4 bw8 | `TAH-Q` fw~4 bw6 |
|---|---|---|---|
| 1Gbps | 2600 | 4749 | 5650 |
| 500Mbps | 2482 | 4311 | 5749 |
| 300Mbps | 1761 | 4369 | 5120 |
| 100Mbps | 751 | 3310 | 4045 |

and (v)), where AQ-SGD becomes infeasible due to prohibitive storage requirements (i.e., Task (iii)) or requiring a single epoch training (i.e., Tasks (iv) and (v)), TAH-QUANT still delivers convergence results closely matching standard FP16 baseline. This underscores TAH-QUANT's significant advantage of achieving aggressive activation quantization without additional memory overhead, making it broadly applicable and scalable in realistic training scenarios. Furthermore, in Table 3, we also report the evaluation results for the SFTed `Qwen2.5-3B` model in Tasks (iii) and (iv), which clearly indicates that models fine-tuned using TAH-QUANT achieve nearly identical downstream task performance compared to the FP16 baseline across multiple benchmarks. This demonstrates that TAH-QUANT effectively maintains model quality while significantly reducing activation communication overhead.

**End-to-end training time.** We show the end-to-end runtime of different methods under slow networks. As illustrated in Figure 3, TAH-QUANT achieves up to 4.3× end-to-end speed-up compared with that of FP16 (in terms of time to the same loss), illustrating the importance of communication compression in slow networks. We also witness a visible speedup when comparing TAH-QUANT with AQ-SGD, where we speculate the speedup is due to the elimination of offloading overhead in AQ-SGD to implement its error compensation mechanism. We summarize the training throughput for the `GPT-2XL` in Table 1 over various slow network conditions, i.e., 1Gbps. 500Mbps, 300Mbps, 100Mbps.

Table 3: `Qwen2.5-3B` SFT evaluation on question-answering and code benchmarks including ARC Clark et al. (2018), TruthfulQA Lin et al. (2022), WinoGrande Sakaguchi et al. (2021), HumanEval Chen et al. (2021b). All evaluations are conducted in zero-shot setting by default. We employ the evaluation scripts from lm-evaluation-harness Gao et al. (2024) framework. For our evaluation, we report: normalized accuracy for ARC-Challenge, accuracy for WinoGrande, mc2 for Truthful QA, and pass@1 scores for Humaneval.

| Model | AVG | Open-Platyups | | | Magicoder-110K |
| | | ARC | TruthfulQA | WinoGrande | HumanEval |
|---|---|---|---|---|---|
| Origin-Qwen2.5 | 51.13 | 47.35 | 48.85 | 68.67 | 39.63 |
| SFT-FP16 | 59.08 | 50.00 | 50.49 | 69.38 | 66.46 |
| SFT-TAH-QUANT | 59.32 | 49.91 | 49.61 | 70.00 | 67.68 |

## 5.3 ABLATION STUDY

To evaluate the specific contributions of each module in TAH-QUANT to limit quantization error in pipeline parallel training, we conduct a series of ablation experiments and enumerate the ablation and experimental results we find below:

Firstly, to study how the **tile-wise quantization group size** influences the statistical efficiency, we vary the group size to 8, 32, and 128 and compare SFTed `Qwen2.5-3B` models over the set of benchmarks. Table 2 illustrates the results — we find that setting up an effective quantization group size affects the final results, when the size is large (i.e., 128), it could affect the quality of the trained models.

Secondly, to examine the effectiveness of the **entropy-guided adaptive bit allocation**, we compare TAH-QUANT with adaptive bit allocation enabled against a variant without adaptive allocation. Results in Figure 4 demonstrate that adaptive allocation accelerates training convergence — the training loss consistently decreases faster with adaptive bit allocation, reflecting reduced quantization error during compression. These observations validate our design choice of incorporating entropy-based token-level bit-width allocation.

Thirdly, We further evaluate the necessity of the **Hadamard-based outlier suppression** component in TAH-QUANT. Comparing training performance between setups with and without the Hadamard transform reveals that including this transform notably improves training stability and convergence speed — in Figure 5, the training loss is substantially lower across the training phase when the Hadamard transform is applied, underscoring its effectiveness in mitigating quantization-induced errors from outlier activations. This finding confirms the value of our pivot element swapping combined with the Hadamard transform in enhancing quantization robustness.

Table 2: Ablation study about tile-wise quantization group size in TAH-QUANT.

| Tile-size | MMLU | ARC |
|---|---|---|
| 8 | 64.60 | 50.34 |
| 32 | 64.88 | 49.91 |
| 128 | 64.34 | 49.66 |

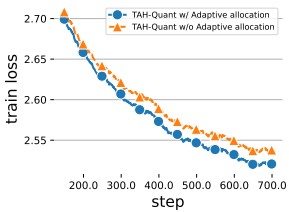

Figure 4: Ablation study for adaptive bit allocation.

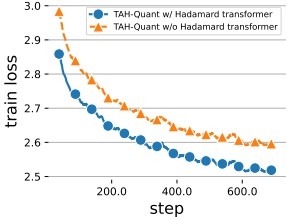

Figure 5: Ablation study for Hadamard transform.

## 6 CONCLUSION

In this paper, we present TAH-QUANT, a novel activation quantization method that alleviates communication bottlenecks in decentralized pipeline-parallel training of LLMs. TAH-QUANT integrates fine-grained tile-wise quantization for localized error control, entropy-guided token-level bit allocation for efficient utilization, and a Hadamard-based transform with pivot swapping to mitigate outliers. We theoretically show that pipeline-parallel training with TAH-QUANT preserves the same convergence rate ($\mathcal{O}(1/\sqrt{T})$) as standard SGD. Empirical results further demonstrate that TAH-QUANT compresses activations to 3–4 bits without degrading convergence, while matching or surpassing state-of-the-art methods such as AQ-SGD, avoiding memory overhead, and maintaining robust generalization across diverse training scenarios.

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

## A   THE USE OF LLMs IN WRITING

We used LLM, namely OPENAI-GPT5, to polish the writing of this manuscript. No other generative AI functionality is used in the writing of this submission.

## B   LIMITATIONS

**Pretraining scale.** Due to limited computational resources and time constraints, our evaluation of the pretraining task is restricted to the early-stage convergence behavior of `Qwen2.5-3B` on the `C4` common crawl corpus. The full potential of TAH-QUANT under prolonged pretraining across diverse datasets and larger model scales remains unexplored. We plan to explore if it is possible to pre-train an LLM from scratch, and go through all of the pretraining corpus following the scaling law in a decentralized environment equipped with TAH-QUANT as an interesting future work.

## C   EXPERIMENTAL DETAILS.

**Fine-tuning.** We fine-tune the `GPT2-xl` on `WikiText-2` and `ArXiv21` for 10 epochs. Specifically, we set the learning rate to 5.0e-6, the batch size to 32 and micro-batch size to 1, max sequence length to 1024 for both datasets. The learning rate decays linearly after warm-up stage.

**Instruction-tuning.** We perform instruction tuning on `Qwen2.5-3B` using `Open-Platyups` and `Magicoder-110K` for 1 and 2 epochs, respectively. The learning rate is set to 2.0e-5, with a batch size of 32 for both datasets. We use a cosine learning rate scheduler for `Open-Platyups`, and a cosine scheduler with a minimum learning rate of 2.0e-6 for `Magicoder-110K`.

**Pre-training.** We pretrain the `Qwen2.5-3B` on the `C4` common crawl corpus for 6000 iterations, with a batch size of 131072 tokens. The learning rate is set to 3.0e-4, and we use a cosine scheduler with a minimum learning rate of 3.0e-5. The weight decay is set to 0.01.

## D   MISSING PROOFS

In this section, we provide detailed proofs for Theorem 4.1. We first prove the following lemma.

**Lemma D.1** (Descent lemma). *Under Assumption 2 and the update rule 6 , it holds that*

$$f(\mathbf{x}^{t+1}) \leq f(\mathbf{x}^t) - \frac{\eta}{2}\|\nabla f(\mathbf{x}^t)\|_2^2 - \left(\frac{1}{2\eta} - \frac{L}{2}\right)\|\mathbf{x}^{t+1} - \mathbf{x}^t\|_2^2 + \frac{\eta}{2}\|\mathbf{m}^t - \nabla f(\mathbf{x}^t)\|_2^2. \quad (10)$$

*Proof.* by Assumption 2 we have

$$f(\mathbf{x}^{t+1}) \leq f(\mathbf{x}^t) + \eta\langle\nabla f(\mathbf{x}^t), \frac{1}{\eta}(\mathbf{x}^{t+1} - \mathbf{x}^t)\rangle + \frac{L}{2}\|\mathbf{x}^{t+1} - \mathbf{x}^t\|_2^2$$

$$= f(\mathbf{x}^t) - \frac{\eta}{2}\|\nabla f(\mathbf{x}^t)\|_2^2 - \frac{1}{2\eta}\|\mathbf{x}^{t+1} - \mathbf{x}^t\|_2^2 + \frac{\eta}{2}\|\nabla f(\mathbf{x}^t) - \mathbf{m}^t\|_2^2 + \frac{L}{2}\|\mathbf{x}^{t+1} - \mathbf{x}^t\|_2^2. \quad (11)$$

where the second equality uses $2\langle a, b\rangle = \|a\|_2^2 + \|b\|_2^2 - \|a - b\|_2^2$   □

**Lemma D.2** (momentum contraction). *Under Assumptions 1-4, if $\delta \in (0, 1)$, it holds that*

$$\mathbb{E}[\|\mathbf{m}^t - \nabla f(\mathbf{x}^t)\|_2^2] \leq \left(1 - \beta_1\left(1 - \frac{\delta}{2}\right)\right)\mathbb{E}[\|\mathbf{m}^{t-1} - \nabla f(\mathbf{x}^{t-1})\|_2^2] + \frac{2L^2}{\delta\beta_1}\mathbb{E}[\|\mathbf{x}^t - \mathbf{x}^{t-1}\|_2^2]$$

$$+ (\beta_1 + 6\beta_1^2)(1 - \delta)\mathbb{E}[\|\nabla f(\mathbf{x}^t)\|_2^2] + 3(2 - \delta)\beta_1^2\sigma^2. \quad (12)$$

*Proof.* According to the update of momentum6, we have

$$\mathbf{m}^t - \nabla f(\mathbf{x}^t) = (1 - \beta_1)(\mathbf{m}^{t-1} - \nabla f(\mathbf{x}^{t-1}) + \nabla f(\mathbf{x}^{t-1}) - \nabla f(\mathbf{x}^t)) + \beta_1(\hat{\mathbf{g}}^t - \nabla f(\mathbf{x}^t)).$$

Taking expectation we have

$$\mathbb{E}[\|\mathbf{m}^t - \nabla f(\mathbf{x}^t)\|_2^2] = \mathbb{E}[\|(1-\beta_1)(\mathbf{m}^{t-1} - \nabla f(\mathbf{x}^{t-1}) + \nabla f(\mathbf{x}^{t-1}) - \nabla f(\mathbf{x}^t)) + \beta_1(\mathbb{E}[\hat{\mathbf{g}}^t] - \nabla f(\mathbf{x}^t))\|_2^2]$$
$$+ \beta_1^2 \mathbb{E}[\|\hat{\mathbf{g}}^t - \mathbb{E}[\hat{\mathbf{g}}^t]\|_2^2]. \tag{13}$$

For the first term, applying Jensen's inequality yields

$$\mathbb{E}[\|(1-\beta_1)(\mathbf{m}^{t-1} - \nabla f(\mathbf{x}^{t-1}) + \nabla f(\mathbf{x}^{t-1}) - \nabla f(\mathbf{x}^t) + \beta_1(\mathbb{E}[\hat{\mathbf{g}}^t] - \nabla f(\mathbf{x}^t))\|_2^2]$$
$$\leq (1-\beta_1)\mathbb{E}[\|\mathbf{m}^{t-1} - \nabla f(\mathbf{x}^{t-1}) + \nabla f(\mathbf{x}^{t-1}) - \nabla f(\mathbf{x}^t)\|_2^2] + \beta_1\mathbb{E}[\|\mathbb{E}[\hat{\mathbf{g}}^t] - \nabla f(\mathbf{x}^t)\|_2^2]. \tag{14}$$

By Young's inequality, we have

$$\mathbb{E}[\|\mathbf{m}^{t-1} - \nabla f(\mathbf{x}^{t-1}) + \nabla f(\mathbf{x}^{t-1}) - \nabla f(\mathbf{x}^t)\|_2^2] \leq \left(1 + \frac{\delta\beta_1}{2}\right)\mathbb{E}[\|\mathbf{m}^{t-1} - \nabla f(\mathbf{x}^{t-1})\|_2^2]$$
$$+ \left(1 + \frac{2}{\delta\beta_1}\right)\mathbb{E}[\|\nabla f(\mathbf{x}^t) - \nabla f(\mathbf{x}^{t-1})\|_2^2]. \tag{15}$$

For the second term, applying Cauchy's inequality yields

$$\mathbb{E}[\|\hat{\mathbf{g}}^t - \mathbb{E}[\hat{\mathbf{g}}^t]\|_2^2] \leq 3\mathbb{E}\|\hat{\mathbf{g}}^t - \mathbf{g}^t\|_2^2 + 3\mathbb{E}[\|\mathbf{g}^t - \nabla f(\mathbf{x}^t)\|_2^2] + 3\mathbb{E}[\|\nabla f(\mathbf{x}^t) - \mathbb{E}[\hat{\mathbf{g}}^t]\|_2^2]$$
$$\leq 3(1-\delta)\mathbb{E}[\|\nabla f(\mathbf{x}^t)\|_2^2] + 3(1-\delta)\mathbb{E}[\|\mathbf{g}^t\|_2^2] + 3\sigma^2,$$
$$\leq 6(1-\delta)\mathbb{E}[\|\nabla f(\mathbf{x}^t)\|_2^2] + 3(2-\delta)\sigma^2, \tag{16}$$

where the inequality uses Assumption 3 and 4. Applying (14)(15)(16) to (13) and using Assumption 2 and 4, we obtainD.2 □

**Remark.** From this proof, it is evident that both inequalities in Assumption 4 are necessary. In particular, the second inequality is essential for bounding the variance of $\hat{\mathbf{g}}^t$, which plays a crucial role in the overall convergence analysis

Now we are ready to prove Theorem 4.1. We first restate the theorem in Theorem D.3.

**Theorem D.3.** *Under Assumptions 1-4, if $\beta_1 \in (0, \delta/(24 - 12\delta)), \delta_1 \in (0,1)$ and $\eta \leq \min\{1/2L, \sqrt{(\delta\beta_1^2)/(8L^2)}\}$, TAH-QUANT with momentum SGD converges as*

$$\frac{1}{T+1}\sum_{t=0}^{T}\mathbb{E}[\|\nabla f(\mathbf{x}^t)\|_2^2] \leq \frac{8[f(\mathbf{x}^0) - \inf_{\mathbf{x}} f(\mathbf{x})]}{\delta\eta(T+1)} + \frac{8\|\mathbf{m}^0 - \nabla f(\mathbf{x}^0)\|_2^2}{\delta\beta_1(T+1)} + \frac{24\beta_1\sigma^2}{\delta}. \tag{17}$$

*Proof.* By Lemma D.1, we have

$$f(\mathbf{x}^{t+1}) - f(\mathbf{x}^t) \leq -\left(\frac{1}{2\eta} - \frac{L}{2}\right)\|\mathbf{x}^{t+1} - \mathbf{x}^t\|_2^2 + \frac{\eta}{2}\|\nabla f(\mathbf{x}^t) - \mathbf{m}^t\|_2^2 - \frac{\eta}{2}\|\nabla f(\mathbf{x}^t)\|_2^2. \tag{18}$$

Taking expectation and summing (18) for $t = 0, 1, \cdots, T$ yields

$$\inf_{\mathbf{x}} f(\mathbf{x}) - f(\mathbf{x}^0) \leq \frac{\eta}{2}\sum_{t=0}^{T}\mathbb{E}[\|\nabla f(\mathbf{x}^t) - \mathbf{m}^t\|_2^2] - \left(\frac{1}{2\eta} - \frac{L}{2}\right)\sum_{t=0}^{T}\mathbb{E}[\|\mathbf{x}^{t+1} - \mathbf{x}^t\|_2^2]$$
$$- \frac{\eta}{2}\sum_{t=0}^{T}\mathbb{E}[\|\nabla f(\mathbf{x}^t)\|_2^2]. \tag{19}$$

summing the inequality in Lemma D.2 for $t = 1, 2, \cdots, T$ we have

$$\beta_1\left(1 - \frac{\delta}{2}\right)\sum_{t=0}^{T}\mathbb{E}[\|\mathbf{m}^t - \nabla f(\mathbf{x}^t)\|_2^2] \leq \|\mathbf{m}^0 - \nabla f(\mathbf{x}^0)\|_2^2 + \frac{2L^2}{\delta\beta_1}\sum_{t=1}^{T}\|\mathbf{x}^t - \mathbf{x}^{t-1}\|_2^2$$
$$+ (1-\delta)(\beta_1 + 6\beta_1^2)\sum_{t=1}^{T}\mathbb{E}[\|\nabla f(\mathbf{x}^t)\|_2^2] + 3T(2-\delta)\beta_1^2\sigma^2. \tag{20}$$

noting that $\delta \in (0, 1)$ we obtain

$$\sum_{t=0}^{T} \mathbb{E}[\|\mathbf{m}^t - \nabla f(\mathbf{x}^t)\|_2^2] \leq \frac{2\|\mathbf{m}^0 - \nabla f(\mathbf{x}^0)\|_2^2}{\beta_1} + \frac{4L^2}{\delta \beta_1^2} \sum_{t=1}^{T} \|\mathbf{x}^t - \mathbf{x}^{t-1}\|_2^2$$

$$+ \left(1 - \frac{\delta}{2}\right)(1 + 6\beta_1) \sum_{t=1}^{T} \mathbb{E}[\|\nabla f(\mathbf{x}^t)\|_2^2] + 6T\beta_1 \sigma^2. \quad (21)$$

Applying 21 to (19) and noting that $\beta_1 \in (0, \delta/(24 - 12\delta))$ implies $(1 - \delta/2)(1 + 6\beta_1) \leq 1 - \delta/4$, we obtain

$$\frac{1}{T+1} \sum_{t=0}^{T} \mathbb{E}[\|\nabla f(\mathbf{x}^t)\|_2^2] \leq \frac{8[f(\mathbf{x}^0) - \inf_{\mathbf{x}} f(\mathbf{x})]}{\delta \eta (T+1)} + \frac{8\|\mathbf{m}^0 - \nabla f(\mathbf{x}^0)\|_2^2}{\delta \beta_1 (T+1)} + \frac{24\beta_1 \sigma^2}{\delta}$$

$$- \frac{8}{\delta \eta} \left(\frac{1}{2\eta} - \frac{L}{2} - \frac{2\eta L^2}{\delta \beta_1^2}\right) \sum_{t=0}^{T} \|\mathbf{x}^{t+1} - \mathbf{x}^t\|_2^2. \quad (22)$$

Since $\eta \leq \min\{1/2L, \sqrt{(\delta \beta_1^2)/(8L^2)}\}$ implies $1/(4\eta) \geq L/2$ and $1/(4\eta) \geq (2\eta L^2)/(\delta \beta_1^2)$, (17) is a direct result of (22). $\qquad \square$

# E    THE USE OF LLMS IN WRITING

We used LLM, namely OPENAI-GPT5, to polish the writing of this manuscript. No other generative AI functionality is used in the writing of this submission.

