# OpenReview forum: "TAH-QUANT: Effective Activation Quantization in Pipeline Parallelism over Slow Network"
_ICLR.cc/2026/Conference — Submitted to ICLR 2026_

### Official Review · Reviewer_dQEq · 2025-10-23

**Soundness:** 2
**Presentation:** 2
**Contribution:** 2
**Rating:** 4
**Confidence:** 3

**Summary:**

The paper proposes a framework to reduce communication costs in decentralized, pipeline-parallel training of large language models. It addresses the bottleneck of transmitting intermediate activations over slow networks through tile-wise quantization (i.e., channel groups). The authors propose entropy-guided adaptive bit allocation for each quantization group, and for groups containing large outliers, they adopt a Hadamard transform with pivot swapping to suppress these outliers and reduce quantization errors. Theoretically, TAH-QUANT is shown to maintain the same convergence rate, O(1/\sqrt{T}), as standard SGD. Empirical studies on GPT-2-XL and Qwen-2.5-3B across multiple datasets demonstrate that it can quantize activations to 3–4 bits, achieving up to 1.33×–4.3× speedups with negligible accuracy loss and no additional memory overhead compared to baselines like AQ-SGD. The method scales efficiently under slow network conditions and generalizes well across both pretraining and instruction-tuning tasks.

**Strengths:**

-  To the best of my knowledge, TAH-QUANT’s combination of tile-wise quantization, entropy-based adaptive bit allocation, and Hadamard outlier suppression is conceptually novel and well-motivated.
- The paper provides a rigorous convergence analysis proving the same O(1/\sqrt{T}) rate as vanilla SGD, as well as showing the practical efficiency and speedup under the limited bandwidth.
- The contributions of each module (tile size, adaptive allocation, Hadamard transform) are clearly validated, strengthening the empirical credibility of the method

**Weaknesses:**

- Although the authors show empirical training speedups under limited bandwidth, the computational costs of the bit allocation, outlier detection heuristic, and Hadamard transform are not discussed.
- The paper also lacks an analysis of memory reduction versus compression cost versus batch size versus bandwidth. I recommend that the authors include a micro-benchmark demonstrating the speedup of transmitting a batch of data while sweeping across different batch sizes and bandwidth settings.
- The analysis of the bit allocation mechanism is missing. This component likely affects both the total number of bytes transferred between machines and the overall training speedup. An ablation study on the top-p% threshold is necessary. Furthermore, the hyperparameter p is not specified (or please point me where it is defined or described in the paper).
- Although Algorithm 1 outlines the overall training process, the detailed procedures for bit allocation, outlier detection, and the Hadamard transform are not presented. I recommend that the authors include a pseudo-code algorithm or a schematic figure to illustrate the detailed workflow of the proposed method.
- The experimental models are relative small. Only Qwen2.5-3B (3B parameters) and GPT-2XL (1.5B parameters) are included.

**Questions:**

- Would the authors provide a discussion on the computational cost of quantization?
- Would the authors include a micro-benchmark evaluating different combinations of batch sizes and bandwidths?
- Could the authors report the memory reduction achieved under different values of top-p%, along with an ablation study analyzing the effect of varying p across different layers and experimental models?
- Would the authors present a detailed description of the proposed quantization and bit-allocation process, perhaps in the form of pseudo-code or a schematic figure?
- If possible, could the authors extend the experiments to larger models to better align with the paper’s discussion on “democratizing large-scale LLM training”?
- How do the authors define “slow network” or “low bandwidth”? Is the 10 Gbps bandwidth used in the paper intended as a formal benchmark from the communication systems domain?

---

> ### Author Response · Authors · 2025-11-24
>
> > W1: Although the authors show empirical training speedups under limited bandwidth, the computational costs of the bit allocation, outlier detection heuristic, and Hadamard transform are not discussed.
> >
>
> > Q1: Would the authors provide a discussion on the computational cost of quantization?
> >
>
> Let $B, S, D, T_{s}$ be the batch size, sequence length , hidden dimension and tile size, respectively.
>
> - Pivot swapping involves finding the top-2 per tile and performing a conditional element swap. This operation has complexity $\mathbb{O}(T_{s})$ per tile, and the swapping operation is implemented using efficient tensor-level operations that are applied to tiles in parallel.
> - Hadamard transformation involves matrix multiplication between a Hadamard matrix and each tile. Its computational complexity is $\mathbb{O}(B\times S \times D \times T_{s})$.
> - The entropy computation scales with $\mathbb{O}(B \times S \times D).$
> - Token-level ranking operation introduces only $\mathbb{O}(S \times \log{(S)})$ complexity.
>
> By contrast, major components in training, such as attention and feedforward layers, incur much higher costs:
>
> - FFN: $\mathbb{O}(B \times S \times D^{2})$.
> - Attention: $\mathbb{O}(B \times S^{2} \times D)$.
>
> We also conducted runtime profiling in a practical setting (4-stage pipeline, micro-batch size 2) and observed that the overall added overhead by TAH-Quant is approximately **0.76%** cost relative to the original training workload.
>
> We hope these clarifications address the reviewer’s concerns.

---

> ### Author Response · Authors · 2025-11-24
>
> > W2: The paper also lacks an analysis of memory reduction versus compression cost versus batch size versus bandwidth. I recommend that the authors include a micro-benchmark demonstrating the speedup of transmitting a batch of data while sweeping across different batch sizes and bandwidth settings.
> >
>
> > Q2: Would the authors include a micro-benchmark evaluating different combinations of batch sizes and bandwidths?
> >
>
> We appreciate this suggestion. To provide a more complete analysis, we will conduct additional micro-benchmarks that systematically sweep across batch sizes and bandwidth settings, measuring end-to-end transmission speedup. We will release these results promptly.

---

> ### Author Response · Authors · 2025-11-24
>
> > W3: The analysis of the bit allocation mechanism is missing. This component likely affects both the total number of bytes transferred between machines and the overall training speedup. An ablation study on the top-p% threshold is necessary. Furthermore, the hyperparameter p is not specified (or please point me where it is defined or described in the paper).
> >
>
> > Q3: Could the authors report the memory reduction achieved under different values of top-p%, along with an ablation study analyzing the effect of varying p across different layers and experimental models?
> >
>
> We thank the reviewer for raising this important point regarding the analysis of the bit-allocation mechanism and the top-p% hyper-parameter.
>
> We use 80% INT4 + 20% INT3 as default configuration, as stated in the Empirical justification of Assumption 4.
>
> **(1) Ablation on different top-p% on INT3 / INT4 allocations.** We additionally evaluated multiple quantization configurations by varying the INT3/INT4 top-p%. The results are shown below:
>
> | step / loss | 50 | 100 | 150 | 200 | 300 | 400 | 500 | 600 |
> | --- | --- | --- | --- | --- | --- | --- | --- | --- |
> | 100% INT4 | 2.78 | 2.70 | 2.66 | 2.61 | 2.56 | 2.52 | 2.51 | 2.49 |
> | 50% INT4 + 50% INT3 | 2.82 | 2.74 | 2.71 | 2.66 | 2.62 | 2.58 | 2.57 | 2.55 |
> | 100% INT3 | 2.85 | 2.77 | 2.74 | 2.69 | 2.65 | 2.62 | 2.60 | 2.58 |
>
> These results indicate that higher-precision bit allocation (INT4) yields more stable convergence, and mixed INT3/INT4 configurations fall between the two extremes. This shows that TAH-Quant’s convergence is sensitive but well-behaved with respect to bit-allocation choices.
>
> Overall, under a fixed bandwidth budget, these experiments demonstrate that practitioners can empirically trade off communication cost and convergence stability by selecting appropriate INT3/INT4 splits or tile-wise bit allocations.
>
> **(2) Memory reduction vs. top-p%.** We agree that bit-allocation affects communication volume, but we want to clarify that it does not meaningfully affect memory footprint. TAH-Quant performs on-the-fly quantization of activations at each pipeline boundary. No activation is cached from previous steps or previous epochs. Therefore, memory usage is dictated by model states and per-layer activations, not by how many tokens are assigned INT3 vs. INT4. In contrast, AQ-SGD must store activations for all tokens across all pipeline stages for the entire dataset, causing memory usage to scale linearly with dataset size × model dimension. TAH-Quant eliminates this cost entirely.

---

> ### Author Response · Authors · 2025-11-24
>
> > W4: Although Algorithm 1 outlines the overall training process, the detailed procedures for bit allocation, outlier detection, and the Hadamard transform are not presented. I recommend that the authors include a pseudo-code algorithm or a schematic figure to illustrate the detailed workflow of the proposed method.
> >
>
> > Q4: Would the authors present a detailed description of the proposed quantization and bit-allocation process, perhaps in the form of pseudo-code or a schematic figure?
> >
>
> We appreciate the reviewer’s meaningful suggestion. In response, we now provide a clearer and more structured description of the proposed quantization workflow.  We will further present a pseudo-code version in the revised paper.
>
> **Activation Tensor Shape and Setup.**
>
> The activation tensor has shape $[B, S, C]$, where: $B$ is the batch size, $S$ is the sequence length (number of tokens),  $C$ is the number of channels (the model dimension). Each token corresponds to a vector of size $C$.
>
> **Quantization Workflow.**
>
> **Step 1: Token-Level Entropy-Guided Bit Allocation (Section 3.2)**
>
> We first analyze each token's activation vector $a_{i,j} \in \mathbb{R}^{C}$ to determine its quantization bit width.
>
> - Compute an entropy score $H(a_{i,j})$  using the L1-normalized magnitudes of its channel-wise values.
> - Based on the entropy ranking across all tokens, we assign higher bit width (e.g., INT4) to the top-p% high-entropy tokens, and lower bit width (e.g., INT3) to the rest.
> - This per-token bit width is then used for all subsequent quantization groups (tiles) derived from the same token.
>
> **Step 2: Tile-Wise Partitioning and Outlier Detection (Section 3.1 + 3.3)**
>
> Next, we divide each token's channel vector along the $C$ dimension into multiple tiles of a certen size $G$ (e.g., 32), yielding $N_{t}$ groups per token:
>
> - Each tile becomes a quantization group $a_{i,j,t} \in \mathbb{R}^{G}$
> - If one group is classified with outlier, we apply the Hadamard-based transform to that tile before quantization. Otherwise, we proceed without the transform.
> - For each group, we check for outliers
>
> **Step 3: Tile-Wise Asymmetric Quantization (Section 3.1 + 3.3)**
>
> Finally, we perform asymmetric uniform quantization on each tile:
>
> - Each tile is quantized independently using its own scale and zero-point.
> - The number of quantization bits per tile is inherited from its parent token (from Step 1).
> - This fine-grained tile-wise quantization, coupled with local dynamic range fitting and selective Hadamard transformation, ensures robustness even under low-bit precision.
>
> **Example:**
>
> Let’s consider a toy example with a 8-element activation vector containing an outlier:
>
> $$
> t = [10, 1.0, -2.0, 0.5, 3.0, -1.0, -1000, 0.9]
> $$
>
> First, we perform a pivot element swap to move the largest absolute value to the first position:
>
> $$
> t^{'} = [-1000, 1.0, -2.0, 0.5, 3.0, -1.0, 10, 0.9]
> $$
>
> Then we multiply it by a 8*8 Hadamard matrix yeilds:
>
> $$
> t^{''} = t^{'} \cdot H = [-349.17, -350.16, -355.82, -354.83, -358.29, -359.42, -349.52, -351.22]
> $$
>
> Next we compress the transformed vector using asymmetric uniform quantization:
>
> $$
> Quant(t^{''}) = [64, 58, 23, 29, 7, 0, 62, 51]
> $$
>
> Finally (in the second machine), to recover the original vector approximately, we apply the inverse Hadamard transform followed by swapping the pivot element back to its original position and get:
>
> $$
> \bar{t} = Q^{-1}(Q(t^{''})) = [10.01, 1.036,-1.956, 0.460, 2.992, -1.036, -1000, 0.921]
> $$
>
> This demonstrates how the Hadamard transform suppresses the outlier and produces a well-behaved distribution suitable for low-bit quantization.

---

> ### Author Response · Authors · 2025-11-24
>
> > W5: The experimental models are relative small. Only Qwen2.5-3B (3B parameters) and GPT-2XL (1.5B parameters) are included.
> >
>
> > Q5: If possible, could the authors extend the experiments to larger models to better align with the paper’s discussion on “democratizing large-scale LLM training”?
> >
>
> We thank the reviewer for the suggestion. We agree that extending the experiments to larger model scales will strengthen the evaluation and better support the paper’s motivation of *democratizing large-scale LLM training*. We will extend additional experiments and release these results as soon as possible.

---

> ### Author Response · Authors · 2025-11-24
>
> > Q6: How do the authors define “slow network” or “low bandwidth”? Is the 10 Gbps bandwidth used in the paper intended as a formal benchmark from the communication systems domain?
> >
>
> We thank the reviewer for raising this important point about our bandwidth configuration.
>
> Recent research has extensively explored decentralized training and inference of large models in bandwidth-constrained environments. These scenarios—with bandwidths significantly lower than NVLINK or PCIe—include training across consumer-grade GPUs, edge clusters, academic labs, and geographically distributed regions connected via Wide Area Networks (WANs).
>
> To demonstrate the practicality and relevance of the <1Gbps bandwidth setting, we cite multiple studies [1~9] that also **use <1Gbps settings in their LLM training experiments**, employing both real deployments and simulations.
>
> While our method focuses on training, we note that **similar bandwidth constraints (<1Gbps) are also present during inference deployments**, especially in decentralized or geo-distributed scenarios [10].
>
> In conclusion, our <1Gbps configuration reflects real-world use cases and aligns with substantial recent literature. We hope these clarifications address the reviewer's concern regarding the use of constrained network bandwidth settings in our experiments.
>
> We appreciate the reviewer's feedback and will elaborate on these points further in the revised manuscript.
>
> [1]: HALoS: Hierarchical Asynchronous Local SGD over Slow Networks for Geo-Distributed Large Language Model Training [ICML 2025]
>
> [2]: Accelerating Model Training in Multi-cluster Environments with Consumer-grade GPUs [SIGCOMM 2024]
>
> [3]: CocktailSGD: fine-tuning foundation models over 500mbps networks [ICML 2023]
>
> [4]: How Can We Train Deep Learning Models Across Clouds and Continents? An Experimental Study [VLDB 2024]
>
> [5]: Accelerating Geo-distributed Machine Learning with Network-Aware Adaptive Tree and Auxiliary Route [Transactions on Networking 2024]
>
> [6]: Distributed Inference and Fine-tuning of Large Language Models Over The Internet [Neurips 2023]
>
> [7]: Decentralized Training of Foundation Models in Heterogeneous Environments [neurips 2022]
>
> [8]: Kimad: Adaptive Gradient Compression with Bandwidth Awareness [DistributedML 2023]
>
> [9]: SWARM Parallelism: Training Large Models Can Be Surprisingly Communication-Efficient [ICML 2023]
>
> [10]: Helix: Serving Large Language Models over Heterogeneous GPUs and Network via Max-Flow [ASPLOS 2025]

---

> > ### Comment · Reviewer_dQEq · 2025-11-25
> > **Reply to authors' rebuttal**
> >
> > Thanks for the clarification. The authors’ rebuttal addresses some of my questions and concerns. I recommend that the authors update the paper during the discussion period to incorporate these points and further strengthen the work. However, several key issues still remain unresolved. For instance, a benchmark that examines memory reduction versus compression cost versus batch size versus bandwidth is still missing, making the trade-offs of the proposed algorithm unclear. In addition, experiments on larger models are still absent. Given these unresolved points, I will maintain my current rating, but I remain open to further discussion with the authors.

---

> > > ### Author Response · Authors · 2025-11-29
> > >
> > > > W2: The paper also lacks an analysis of memory reduction versus compression cost versus batch size versus bandwidth. I recommend that the authors include a micro-benchmark demonstrating the speedup of transmitting a batch of data while sweeping across different batch sizes and bandwidth settings.
> > > >
> > >
> > > > Q2: Would the authors include a micro-benchmark evaluating different combinations of batch sizes and bandwidths?
> > > >
> > >
> > > > Feedback: Thanks for the clarification. The authors’ rebuttal addresses some of my questions and concerns. I recommend that the authors update the paper during the discussion period to incorporate these points and further strengthen the work. However, several key issues still remain unresolved. For instance, a benchmark that examines memory reduction versus compression cost versus batch size versus bandwidth is still missing, making the trade-offs of the proposed algorithm unclear. In addition, experiments on larger models are still absent. Given these unresolved points, I will maintain my current rating, but I remain open to further discussion with the authors
> > > >
> > >
> > > We thank the reviewer for the constructive feedback. Following the reviewer’s suggestion, we have now conducted the requested micro-benchmarks that sweep **batch size** and **bandwidth**, measuring end-to-end **throughput (tokens/s)** and **speedup.**
> > >
> > > 1. Micro-benchmark results
> > >     - Global batch = 32, bandwidth = 500 Mbps
> > >
> > >     | throughput (tokens/s) | w/o TAH-Quant | with TAH-Quant | Speedup |
> > >     | --- | --- | --- | --- |
> > >     | mbs = 1 | 2406 | 4289 | **1.78×** |
> > >     | mbs = 2 | 1710 | 4016 | **2.35×** |
> > >     | mbs = 4 | 1317 | 3023 | **2.30×** |
> > >     | mbs = 8 | 981 | 2099 | **2.14×** |
> > >     - Global batch = 32, bandwidth = 1 Gbps
> > >
> > >     | throughput (tokens/s) | w/o TAH-Quant | with TAH-Quant | Speedup |
> > >     | --- | --- | --- | --- |
> > >     | mbs = 1 | 3449 | 5177 | **1.50×** |
> > >     | mbs = 2 | 2576 | 4655 | **1.81×** |
> > >     | mbs = 4 | 2003 | 3475 | **1.73×** |
> > >     | mbs = 8 | 1485 | 2436 | **1.64×** |
> > > 2. Analysis
> > >     - Lower bandwidth → communication bottleneck dominates → TAH-Quant provides larger gains.
> > >     At **500 Mbps**, inter-stage communication becomes the primary bottleneck. Because TAH-Quant aggressively reduces activation size via low-bit quantization, the communication time shrinks substantially, leading to **up to 2.35× speedup**.
> > >     - Larger mbs → more communication per step → larger benefit from compression.
> > >
> > >         Increasing the *micro-batch size* increases tensor volume transmitted across PP stages. As a result, compression benefits also increase, particularly at low bandwidth. The speedup peaks around **mbs=2–4** for both bandwidth settings.
> > >
> > >     - Very large mbs introduces pipeline bubbles → slight decline in speedup.
> > >
> > >         When mbs becomes too large (e.g., mbs = 8), the pipeline depth remains fixed but each micro-batch becomes heavier, creating **pipeline bubbles** and additional idle time. This reduces the relative fraction of communication time in the total iteration cost, leading to a modest reduction in speedup.

---

> > > ### Author Response · Authors · 2025-11-29
> > >
> > > > W5: The experimental models are relative small. Only Qwen2.5-3B (3B parameters) and GPT-2XL (1.5B parameters) are included.
> > > >
> > >
> > > > Q5: If possible, could the authors extend the experiments to larger models to better align with the paper’s discussion on “democratizing large-scale LLM training”?
> > > >
> > >
> > > > Feedback: Thanks for the clarification. The authors’ rebuttal addresses some of my questions and concerns. I recommend that the authors update the paper during the discussion period to incorporate these points and further strengthen the work. However, several key issues still remain unresolved. For instance, a benchmark that examines memory reduction versus compression cost versus batch size versus bandwidth is still missing, making the trade-offs of the proposed algorithm unclear. In addition, experiments on larger models are still absent. Given these unresolved points, I will maintain my current rating, but I remain open to further discussion with the authors
> > > >
> > >
> > > We thank the reviewer for the continued feedback and for highlighting the importance of evaluating the proposed method at larger model scales. Following the reviewer’s suggestion, we have now conducted a new experiment on a substantially larger model: **LLaMA-8B** pretrained on the **OpenWebMath** dataset (14.7B tokens). We trained the model for **8B tokens (≈55% of the dataset)**, which required approximately **600 GPU hours**, under a realistic large-scale training configuration (**sequence length 8K, global batch size 128, pipeline parallelism = 4**). We compare training *with* and *without* TAH-Quant under identical settings.
> > >
> > > The results are summarized below:
> > >
> > > | token num / loss | 1B | 2B | 3B | 4B | 5B | 6B | 7B | 8B |
> > > | --- | --- | --- | --- | --- | --- | --- | --- | --- |
> > > | **with TAH-Quant** | 2.694 | 2.389 | 2.268 | 2.172 | 2.092 | 2.031 | 1.999 | 1.953 |
> > > | **without TAH-Quant** | 2.691 | 2.393 | 2.284 | 2.181 | 2.102 | 2.038 | 2.005 | 1.959 |
> > >
> > > These results demonstrate that **TAH-Quant scales effectively to 8B-parameter models** while preserving training stability and end-to-end convergence. The loss curves of both settings closely match across the entire 8B-token training trajectory, confirming that our proposed compression pipeline remains reliable at larger scales.
> > >
> > > Overall, we hope these new results address the reviewer’s concerns regarding model scale and provide further evidence supporting our claim of *democratizing large-scale LLM training*.

---

### Official Review · Reviewer_W7By · 2025-11-01

**Soundness:** 3
**Presentation:** 3
**Contribution:** 3
**Rating:** 6
**Confidence:** 4

**Summary:**

The paper addresses the communication bottleneck of pipeline-parallel training for LLM by compressing activations. It proposes TAH-QUANT (Tile-wise Adaptive Hadamard Quantization) that has: (i) fine-grained tile-wise quantization, (ii) entropy-guided, token-level adaptive bit allocation, and (iii) a Hadamard transform with pivot element swapping. Unlike error-compensation methods (e.g., AQ-SGD), TAH-QUANT claims no extra memory overhead. The authors also provide a convergence analysis showing comparable rates to vanilla SGD, and report activation compression yielding up to 1.33X end-to-end speedups without harming convergence.

**Strengths:**

- Important novelty: this paper proposed multiple novel techniques in quantizing communication in LLM training, such as tile-wise quantization, entropy-guided bit allocation, Hadamard transform, etc.
 - Good theory: authors proved theoretically TAH-QUANT has similar convergence rate as vanilla SGD.
 - Convincing results: TAH-QUANT has up to 1.33X speedup without hurting convergence.

**Weaknesses:**

- The outlier detection heuristic in Section 3.3 is hand made. Could authors explain how well it generalizes to other training settings?
 - More experiments may be needed:
   - The 3090 GPU is not a powerful machine. How will the performance be if we conduct experiments on industrial computation resources, such as A100, H100, H200?
   - Why do we use 8 pipeline parallel stages on 8 GPUs? In practice, we may not need so many PP stages. If we set the number of PP stages to be smaller than the number of GPUs, for example, on 8 GPUs, if we set PP=2, how will the performance be?

**Questions:**

- Can we use the proposed TAH-QUANT method in other training techniques, in addition to pipeline parallel? For example, can we extend it to tensor parallel, data parallel, expert parallel?

---

> ### Author Response · Authors · 2025-11-24
>
> > W1: The outlier detection heuristic in Section 3.3 is hand made. Could authors explain how well it generalizes to other training settings?
> >
>
> We appreciate the reviewer’s feedback. To further assess the robustness of our outlier detection method, we will conduct additional ablation studies on different threshold values and report the results promptly.
>
> We would like to emphasize that our outlier-detection mechanism is a principled component designed to work together with pivot swapping, Hadamard-based outlier suppression, and tile-wise quantization. These components are jointly constructed to address the fundamental challenge of activation compression in pipeline-parallel training. While our experiments already cover three training scenarios (pre-training, fine-tuning, instruction tuning) across two models, we acknowledge that the chosen threshold is based on empirical observations. To assess the robustness of our design, we performed ablations by varying the threshold parameter p. The results (shown below) demonstrate that the proposed detection mechanism consistently improves convergence stability, and empirically we find that a threshold of p=2 works well:
>
> | loss | step 100 | step 200 | step 300 | step 400 | step 500 |
> | --- | --- | --- | --- | --- | --- |
> | p = 0 (assume every tile has outliers) | 2.97 | 2.79 | 2.63 | 2.60 | 2.60 |
> | **p = 2 (ours)** | **2.72** | **2.63** | **2.59** | **2.56** | **2.55** |
> | p = 4 | 2.73 | 2.65 | 2.61 | 2.58 | 2.57 |
> | p = +inf (assume no tile has outliers) | 2.82 | 2.72 | 2.68 | 2.64 | 2.63 |
>
> These results confirm that the outlier detection module plays a meaningful role in stabilizing low-bit activation quantization.

---

> ### Author Response · Authors · 2025-11-24
>
> > W2: More experiments may be needed:
> • The 3090 GPU is not a powerful machine. How will the performance be if we conduct experiments on industrial computation resources, such as A100, H100, H200?
> • Why do we use 8 pipeline parallel stages on 8 GPUs? In practice, we may not need so many PP stages. If we set the number of PP stages to be smaller than the number of GPUs, for example, on 8 GPUs, if we set PP=2, how will the performance be?
> >
>
> We thank the reviewer for the helpful suggestions. Below, we clarify both concerns.
>
> **(1) Why are experiments conducted on 3090 GPUs instead of A100/H100/H200?**
>
> Our target scenario is decentralized training, where computation is contributed by geographically distributed participants using heterogeneous and often resource-limited hardware. In this setting, consumer-grade GPUs such as the RTX 3090 are the realistic deployment environment, and communication bottlenecks in pipeline-parallel training become significantly more challenging. Evaluating TAH-Quant under these constraints directly aligns with our goal: enabling efficient training when both network bandwidth and GPU memory are limited. We acknowledge that results on A100/H100/H200 would be valuable, but they correspond to a different use case (intra-datacenter training rather than decentralized training), and are outside the scope of this work.
>
> **(2) Why use PP=8 on 8 GPUs instead of PP=2.**
>
> Given the limited memory of 3090 GPUs, reducing PP to 2 would force each stage to host a much larger portion of the model, which would either (i) exceed memory capacity, (ii) require shrinking the batch size, or (iii) require CPU offloading—each of which would slow training. In addition, with PP=2, one must introduce other strategies, such as data parallelism to utilize all 8 GPUs, which adds substantial extra communication under decentralized network conditions. For these reasons, PP=8 on 3090s reflects a practical and realistic decentralized setup. Extending to other hardware and parallelization configurations is promising future work, but beyond the present study.

---

> ### Author Response · Authors · 2025-11-24
>
> > Q1: Can we use the proposed TAH-QUANT method in other training techniques, in addition to pipeline parallel? For example, can we extend it to tensor parallel, data parallel, expert parallel?
> >
>
> We thank the reviewer for this interesting question. Conceptually, TAH-Quant is not restricted to pipeline parallelism, because its core mechanism—on-the-fly activation quantization—can be applied to any distributed strategy where activations are communicated across devices.
>
> - **Tensor Parallel & Expert Parallel.** In both tensor parallel (TP) and expert parallel (EP) training, activations are exchanged across GPUs. Since TAH-Quant operates directly on activations without requiring cached states, it can be naturally integrated into these settings.
> - **Data Parallel.** Data parallelism is different because the main communication object is model parameters, not activations. Extending TAH-Quant to DP would therefore require adapting the Hadamard–based quantization to weight/gradient tensors. A practical challenge is that the Hadamard transform requires the tile size to be a power of two, while weight matrices often do not align with such tile boundaries. One potential solution is to pad tensors to the nearest tile size before applying the transform, though this introduces extra memory and computation overhead. Exploring such design choices is an interesting direction for future work.

---

### Official Review · Reviewer_Nj5V · 2025-11-01

**Soundness:** 3
**Presentation:** 2
**Contribution:** 2
**Rating:** 4
**Confidence:** 2

**Summary:**

This paper tackles the challenge of activation communication bottlenecks in pipeline-parallel LLM training, especially over slow or low-bandwidth networks. The authors introduce TAH-QUANT, a quantization framework that compresses activations more efficiently without hurting convergence. It combines three key ideas: (1) tile-wise group quantization for localized precision control, (2) entropy-guided adaptive bit allocation that assigns either 3 or 4 bits per token based on its information content, and (3) a Hadamard transform with pivot swapping to spread outliers and reduce quantization error.

The paper also includes a theoretical convergence analysis under mild assumptions, showing that TAH-QUANT retains the same
√T  convergence rate as standard SGD. Experiments on GPT-XLand Qwen2.5 (for both fine-tuning and short pretraining) show that TAH-QUANT matches FP16 and AQ-SGD in accuracy while achieving substantial end-to-end speedups

**Strengths:**

TAH-QUANT directly tackles a practical and underexplored challenge: activation communication in pipeline-parallel training over limited-bandwidth networks. The method is conceptually simple but well thought out. The combination of tile-wise quantization, adaptive bit allocation, and a Hadamard transform with pivot swapping is elegant and easy to integrate into existing systems.

The theoretical analysis is a strong point. The authors provide √T  convergence guarantees, showing that the method maintains the same rate as standard SGD. This is a meaningful theoretical contribution that helps justify the approach beyond empirical performance.

**Weaknesses:**

The comparison between TAH-QUANT and AQ-SGD feels incomplete. While TAH-QUANT is meant to serve as a more efficient replacement, the experiments don’t fully explore how the two methods compare on broader benchmarks or zero-shot evaluations. Without this, it’s hard to tell whether TAH-QUANT consistently matches or outperforms AQ-SGD in downstream performance.

The loss curves in Figure 2 also don’t add much clarity. They show that both methods converge, but not how or why their trajectories might differ. If the losses are nearly identical, that deserves more explanation. What exactly enables TAH-QUANT to behave so similarly to AQ-SGD, given the absence of error compensation?

It would also help to include additional quantization settings, like loss curves with the different tile sizes or bit allocation ratios, to show whether TAH-QUANT’s behavior holds across a range of precision levels.

**Questions:**

1. Can you include zero-shot evaluation results across more benchmarks to make the comparison with AQ-SGD more convincing?

2. If the loss trajectories between TAH-QUANT and AQ-SGD are almost identical, what’s the underlying reason? Does the Hadamard transform or adaptive bit allocation implicitly mimic error feedback?

3. Have you tested multiple quantization configurations (for example, varying INT3/INT4 splits or tile sizes) to show how sensitive convergence is to these parameters?

4. Can you clarify whether the observed speedups mainly come from algorithmic efficiency or implementation differences compared to AQ-SGD?

---

> ### Author Response · Authors · 2025-11-24
>
> > Q1: Can you include zero-shot evaluation results across more benchmarks to make the comparison with AQ-SGD more convincing?
> >
>
> We thank the reviewer for this helpful suggestion. We would like to clarify an important constraint of AQ-SGD: the method depends on cached activations from previous epochs to compute its error-feedback signal. Therefore, AQ-SGD cannot operate in training settings where only a *single* SFT epoch is used—as in our experimental setup—and consequently cannot be evaluated under the corresponding zero-shot evaluation protocol.
>
> In contrast, TAH-Quant performs fully on-the-fly quantization without requiring any cross-epoch state, making it compatible with single-epoch SFT and the subsequent zero-shot evaluation.

---

> ### Author Response · Authors · 2025-11-24
>
> > Q2: If the loss trajectories between TAH-QUANT and AQ-SGD are almost identical, what’s the underlying reason? Does the Hadamard transform or adaptive bit allocation implicitly mimic error feedback?
> >
>
> We appreciate the reviewer’s question. Although the loss curves of TAH-Quant and AQ-SGD appear similar, the underlying reasons are fundamentally different.
>
> - **AQ-SGD relies on explicit error feedback**, which requires caching activations from previous epochs. Its error signal is computed by comparing the same sample across consecutive epochs, making the method applicable only in multi-epoch training on the same dataset.
> - **TAH-Quant performs on-the-fly quantization with zero activation storage**, and its stability does not arise from implicitly mimicking error feedback. Instead, TAH-Quant’s robustness comes from its three principled design components: Hadamard-based outlier suppression, token-level adaptive bit allocation, and fine-grained tile-wise group quantization. These components substantially reduce quantization error within each step of forward and backward propagation.
>
> These mechanisms eliminate error accumulation at the source, rather than caching and correcting errors across epochs as in AQ-SGD. Thus, even if the two methods exhibit similar loss trajectories, TAH-Quant achieves this through immediate per-step error control, not through delayed error-feedback correction.

---

> ### Author Response · Authors · 2025-11-24
>
> > Q3: Have you tested multiple quantization configurations (for example, varying INT3/INT4 splits or tile sizes) to show how sensitive convergence is to these parameters?
> >
>
> We thank the reviewer for the insightful question. We additionally evaluated multiple quantization configurations by varying the INT3/INT4 ratios. The results are shown below:
>
> | step / loss | 50 | 100 | 150 | 200 | 300 | 400 | 500 | 600 |
> | --- | --- | --- | --- | --- | --- | --- | --- | --- |
> | 100% INT4 | 2.78 | 2.70 | 2.66 | 2.61 | 2.56 | 2.52 | 2.51 | 2.49 |
> | 50% INT4 + 50% INT3 | 2.82 | 2.74 | 2.71 | 2.66 | 2.62 | 2.58 | 2.57 | 2.55 |
> | 100% INT3 | 2.85 | 2.77 | 2.74 | 2.69 | 2.65 | 2.62 | 2.60 | 2.58 |
>
> These results indicate that higher-precision bit allocation (INT4) yields more stable convergence, and mixed INT3/INT4 configurations fall between the two extremes. This shows that TAH-Quant’s convergence is sensitive but well-behaved with respect to bit-allocation choices.
>
> Overall, under a fixed bandwidth budget, these experiments demonstrate that practitioners can empirically trade off communication cost and convergence stability by selecting appropriate INT3/INT4 splits or tile-wise bit allocations.

---

> ### Author Response · Authors · 2025-11-24
>
> > Can you clarify whether the observed speedups mainly come from algorithmic efficiency or implementation differences compared to AQ-SGD?
> >
>
> The speedup primarily comes from the *algorithmic design* of TAH-Quant. Although both TAH-Quant and AQ-SGD reduce communication volume, their underlying mechanisms differ fundamentally:
>
> - **Algorithmic efficiency**. TAH-Quant performs mixed 3/4-bit on-the-fly activation quantization, without storing or retrieving any activations. This enables a more aggressive compression strategy and avoids additional storage-device interactions. In contrast, AQ-SGD requires reading the cached activations from the previous epoch and writing the current activations back to disk for future error-feedback computation—introducing non-negligible I/O overhead that slows down training.
> - **Memory usage**. Since TAH-Quant never caches activations, it incurs **zero extra memory cost**. AQ-SGD must store activations for *every token and every pipeline stage* across the entire dataset, and this overhead grows with both model size and data size, affecting speed and scalability.
> - **General applicability**. AQ-SGD depends on cached activations from prior epochs and therefore only works in *multi-epoch* training settings. This makes it unsuitable for large-scale pretraining and single-epoch fine-tuning, commonly used in LLM training. TAH-Quant does not impose this constraint, contributing to its superior practical efficiency.
>
> Overall, the observed **1.33× speedup** is mainly due to TAH-Quant’s on-the-fly, storage-free algorithmic design rather than any implementation-specific optimizations.

---

### Official Review · Reviewer_7EhT · 2025-11-01

**Soundness:** 3
**Presentation:** 3
**Contribution:** 3
**Rating:** 4
**Confidence:** 3

**Summary:**

The paper addresses the communication bottleneck in decentralized large language model training with pipeline parallelism, which is challenge for democratizing LLM training such as enabling participation from universities or startups with slow network connections. The authors propose TAH-QUANT, a tile-wise adaptive Hadamard quantization framework that integrates three core components: (1) fine-grained tile-wise group quantization for localized error control; (2) entropy-guided token-level adaptive bit allocation (3–4 bits) for efficient compression; and (3) Hadamard transform with pivot swapping to suppress quantization outliers.

The paper evaluated across 5 tasks (language modeling, instruction tuning, pre-training) on GPT2-XL (1.5B) and Qwen2.5-3B (3B), TAH-QUANT achieves up to 4.3× end-to-end speedup over FP16 in slow networks

**Strengths:**

1. TAH-QUANT optimizes pipeline parallelism this by eliminating extra memory overhead while maintaining low-bit quantization (3–4 bits), which is  critical design choice for deployment.

2. Unlike many quantization works that rely solely on empirical validation, the paper provides a solid convergence analysis under standard stochastic optimization assumptions.

**Weaknesses:**

1. The experiment is restricted to Qwen2.5-3B (3B parameters) for only 6,000 iterations on C4. It would be better to anaylyze model at different scales (smaller and larger) to analysis of how TAH-QUANT performs when model size or training duration changes.

2. The outlier detection heuristic uses a fixed threshold without justification. Different LLMs (e.g., GPT-2 vs. Qwen) or tasks have distinct activation distributions.

3. The paper uses naive quantization for gradients in the backward pass (instead of TAH-QUANT) and claims this is due to more computation load enabling communication-computation overlapping. However, no experiments compare this choice to using TAH-QUANT for gradients.

**Questions:**

1. Can you provide a breakdown of end-to-end training time to show that the transform’s overhead does not offset communication savings? For example: Does the Hadamard step increase per-batch compute time by <5% , and how does this scale with tile size?


2. The paper only compares TAH-QUANT to AQ-SGD (a 2021 method) as a low-bit activation compression baseline. However, recent works (e.g., 2023–2024) like LightQuant (NeurIPS 2023) or PipeQuant (ICML 2024) also target pipeline parallelism with low memory overhead. Can you provide a qualitative analysis of how TAH-QUANT’s design differs from these methods?

3. Most modern LLM trainings use mixed precision (e.g., FP16 for weights, FP8 for gradients) to reduce memory/compute. The paper trains models in FP16. Could the author address whether TAH-QUANT is compatible with mixed precision?

---

> ### Author Response · Authors · 2025-11-24
>
> > W1: The experiment is restricted to Qwen2.5-3B (3B parameters) for only 6,000 iterations on C4. It would be better to anaylyze model at different scales (smaller and larger) to analysis of how TAH-QUANT performs when model size or training duration changes.
>
> We appreciate the reviewer’s suggestion. We acknowledge that the C4 pre-training experiment covers only a short training duration. To address this concern, we are working on extending the pre-training experiments on the OpenWebMath dataset and will release these additional results as soon as possible.

---

> > ### Author Response · Authors · 2025-11-29
> >
> > > W1: The experiment is restricted to Qwen2.5-3B (3B parameters) for only 6,000 iterations on C4. It would be better to anaylyze model at different scales (smaller and larger) to analysis of how TAH-QUANT performs when model size or training duration changes.
> > >
> >
> > To address this concern, we have conducted a substantially more comprehensive experiment on a **larger model (LLaMA-8B)** and over a **much longer training duration**. Specifically, we pre-trained LLaMA-8B on the **OpenWebMath** dataset for **8B tokens** (≈55% of the full 14.7B-token corpus), which corresponds to a realistic long-horizon training setting and required approximately **600 GPU hours**. This new experiment uses a production-scale configuration (sequence length 8K, global batch size 128, pipeline parallelism = 4).
> >
> > The training curves *with* and *without* TAH-Quant are shown below:
> >
> > | token num / loss | 1B | 2B | 3B | 4B | 5B | 6B | 7B | 8B |
> > | --- | --- | --- | --- | --- | --- | --- | --- | --- |
> > | **with TAH-Quant** | 2.694 | 2.389 | 2.268 | 2.172 | 2.092 | 2.031 | 1.999 | 1.953 |
> > | **without TAH-Quant** | 2.691 | 2.393 | 2.284 | 2.181 | 2.102 | 2.038 | 2.005 | 1.959 |
> >
> > The results demonstrate that **TAH-Quant maintains stable convergence throughout the entire 8B-token training trajectory**, even at significantly larger model scales and training durations. This provides strong supporting evidence that our method generalizes well beyond the shorter 6,000-step C4 experiment.

---

> ### Author Response · Authors · 2025-11-24
>
> > W2: The outlier detection heuristic uses a fixed threshold without justification. Different LLMs (e.g., GPT-2 vs. Qwen) or tasks have distinct activation distributions.
> >
>
> We appreciate the reviewer’s feedback. We want to clarify that this outlier detection method is developed from a principled design explicitly targeting the challenge of activation compression in pipeline parallel training.
>
> We acknowledge that different LLMs may exhibit different activation statistics. To assess the robustness of our design, we performed ablations by varying the threshold parameter p. The results (shown below) demonstrate that the proposed detection mechanism consistently improves convergence stability, and empirically, we find that a threshold of p=2 works well:
>
> | loss | step 100 | step 200 | step 300 | step 400 | step 500 |
> | --- | --- | --- | --- | --- | --- |
> | p = 0 (assume every tile has outliers) | 2.97 | 2.79 | 2.63 | 2.60 | 2.60 |
> | **p = 2 (ours)** | **2.72** | **2.63** | **2.59** | **2.56** | **2.55** |
> | p = 4 | 2.73 | 2.65 | 2.61 | 2.58 | 2.57 |
> | p = +inf (assume no tile has outliers) | 2.82 | 2.72 | 2.68 | 2.64 | 2.63 |
>
> These results confirm that the outlier detection module plays a meaningful role in stabilizing low-bit activation quantization.

---

> ### Author Response · Authors · 2025-11-24
>
> > W3: The paper uses naive quantization for gradients in the backward pass (instead of TAH-QUANT) and claims this is due to more computation load enabling communication-computation overlapping. However, no experiments compare this choice to using TAH-QUANT for gradients.
> >
>
> We want to gently point out that model gradients, unlike activations, have nearly twice as much time available for communication-computation overlap—since backward propagation typically takes about twice the time of forward propagation—making the naive quantization method with higher precision (e.g., 6-bit) effective enough. We also note that AQ-SGD adopts a similarly naive (fixed-point) gradient compression strategy during the backward pass, likely for similar reasons.
>
> In addition, we conducted experiments to address this question. Results show that when using NAIVE-4bit quantization for backward propagation, the model diverges, while using TAH-Quant-4bit leads to convergence. This suggests that when using the same ultra-low precision in backward propagation, TAH-Quant outperforms the NAIVE method. The specific results are as follows:
>
> |  | step 0 | step 100 | step 200 | step 300 | step 400 | step 500 | step 600 |
> | --- | --- | --- | --- | --- | --- | --- | --- |
> | TAH-Quant-4bit | 2.865 | 2.707 | 2.628 | 2.591 | 2.558 | 2.554 | 2.520 |
> | NAIVE-4bit | 2.871 | 3.044 | 3.161 | 3.205 | 3.489 | 4.530 | 6.039 |
>
> We will clearly explain in our revised manuscript that our reasoning for using the Naive compressor during backward propagation, while also highlighting TAH-Quant's superior convergence stability compared to the naive method when both operate at ultra-low precision (4-bit) in backward propagation.

---

> ### Author Response · Authors · 2025-11-24
>
> > Q1: Can you provide a breakdown of end-to-end training time to show that the transform’s overhead does not offset communication savings? For example: Does the Hadamard step increase per-batch compute time by <5% , and how does this scale with tile size?
> >
>
> Let $B, S, D, T_{s}$ be the batch size, sequence length , hidden dimension and tile size, respectively.
>
> - Pivot swapping involves finding the top-2 per tile and performing a conditional element swap. This operation has complexity $\mathbb{O}(T_{s})$ per tile, and the swapping operation is implemented using efficient tensor-level operations that are applied to tiles in parallel.
> - Hadamard transformation involves matrix multiplication between a Hadamard matrix and each tile. Its computational complexity is $\mathbb{O}(B\times S \times D \times T_{s})$.
> - The entropy computation scales with $\mathbb{O}(B \times S \times D).$
> - Token-level ranking operation introduces only $\mathbb{O}(S \times \log{(S)})$ complexity.
>
> By contrast, major components in training, such as attention and feedforward layers, incur much higher costs:
>
> - FFN: $\mathbb{O}(B \times S \times D^{2})$.
> - Attention: $\mathbb{O}(B \times S^{2} \times D)$.
>
> We also conducted runtime profiling in a practical setting (4-stage pipeline, micro-batch size 2) and observed that the overall added overhead by TAH-Quant is approximately **0.76%** cost relative to the original training workload.
>
> We hope these clarifications address the reviewer’s concerns.

---

> ### Author Response · Authors · 2025-11-24
>
> > Q3: Most modern LLM trainings use mixed precision (e.g., FP16 for weights, FP8 for gradients) to reduce memory/compute. The paper trains models in FP16. Could the author address whether TAH-QUANT is compatible with mixed precision?
> >
>
> The short answer is YES. TAH-Quant is developed from a principled design explicitly targeting the challenge of activation compression in pipeline parallel training.
>
> - Forward pass. TAH-Quant uses token-level adaptive bit allocation, combined with Hadamard-based outlier-suppression transformations and tile-wise quantization. These components are designed to aggressively compress activations to ultra-low precision (e.g., 3/4-bit) without compromising model convergence. Since these transformations operate on quantized activations rather than native weights, they do not conflict with the model being trained in FP16.
> - Backwards pass. We adopt simple but sufficiently effective low-bit quantization scheme for backward gradients. This design is compatible with modern mixed-precision pipelines (FP8 gradients).
>
> Overall, TAH-Quant introduces a low-bit activation communication layer orthogonal to the mixed-precision format used for weights and gradients. As a result, TAH-Quant can be applied in FP16/FP8 mixed-precision training workflows.

---

### Author Response · Authors · 2025-12-03
**Summary**

Dear Area Chair,

We sincerely appreciate your time and effort in handling our submission. As the discussion period is coming to an end, we would like to offer a brief summary to ensure you have the full context of our work when making the final decision.

**A Note on the Validity of One Reviewer’s Claims**.

During the discussion phase, we found that Reviewer 7EhT cited two works (“LightQuant, NeurIPS 2023” and “PipeQuant, ICML 2024”) that do not appear to exist. Despite an extensive search, we were unable to find any record of these papers or their authors. We respectfully mention this, as the nonexistent citations suggest the possibility of AI-generated hallucination. Independent third-party AI-content detection tools also indicated a high likelihood that this portion of the review was machine-generated. We leave this entirely to your judgment and sincerely appreciate your understanding.

**Summary of Our Main Contribution**.

TAH-Quant addresses a fundamental challenge in pipeline-parallel large language model training under decentralized settings.

Existing activation compression methods (e.g., AQ-SGD) rely on history-based error-compensation, which introduces substantial activation memory overhead, limit scalability acoss diverse training regimes.

TAH-Quant addresses these limitations through a fully on-the-fly activation quantization framework. It integrates three principled components:

- Tile-wise fine-grained quantization.
- Entropy-guided token-level adaptive bit-width allocation.
- Hadamard transform + Pivot swap for outlier suppression.

Empirically, TAH-Quant achieves aggressive 3-4 bit activation compression and delivers up to 1.33x end-to-end speedup compared to baseline without compromising convergence. We further provide a theoretical analysis proving that TAH-Quant preserves the $\mathbb{O}(\frac{1}{\sqrt{T}})$ convergence guarantee of vanilla SGD.

**Summary of Reviews**.

Reviewer W7By gives an overall rating of 6. He acknowledges the novelty of our design, our theoretical proof, and the practical effectiveness of the proposed algorithm.

Three reviewers (dQEq, Nj5V, 7EhT) give the overall rating of 4. They also recognize the conceptual soundness of our method and value the provided convergence theory. Their primary requests center on additional experiments on larger models, broader micro-benchmarks, sensitivity on bit allocation, and justification of the outlier detection mechanism.

We have conducted new experiments and provided detailed explanations addressing these points during the discussion period.

**Additional Experiments During Discussion Period**.

Below, we summarize the substantial new experiments that directly address the reviewers’ primary concerns. These were completed after the original submission and documented in our discussion posts.

- Large-Scale Experiment on LLaMA-8B ( $\approx$ 600 GPU hours) (Reviewers 7EhT, dQEq). We conducted a long-term experiment on LLaMA-8B, pre-trained for 8B tokens ( $\approx55%$% of OpenWebMath).
- Micro-benchmarks Sweeping Batch Size & Bandwidth (Reviewers dQEq). We performed the micro-benchmark requested by the reviewers: Bandwidth $\in \set{500Mbps, 1Gbps}$; Micro-batch size $\in \set{1, 2, 4, 8}$; End-to-end throughput (tokens/s) and speedup measured.
- Ablations on INT3/INT4 Bit Allocation Ratios (Reviewer Nj5V). We evaluated multiple INT3/INT4 configurations: 100% INT4; 50% INT4 + 50% INT3; 100% INT3;
- Outlier-Detection Threshold Ablation (Reviewers 7EhT, W7By). We swept the outlier-detection threshold p $\in \set{0, 2, 4, +\infty}$.
- Backward-Pass Quantization Comparison (Reviewer 7EhT). We added experiments showing Naive-4bit diverges, whereas TAH-Quant with gradient scaling converges.

**Clarifications During Discussion Period**.

Below summarizes the main clarifications provided during the discussion.

- Detailed Quantization Workflow (Reviewer dQEq). A complete step-by-step explanation of our quantization workflow.
- Computational Overhead Analysis (Reviewers 7EhT, dQEq). We give a theoretical complexity comparison of Quantization vs. Attention/FFN and performed runtime profiling.
- Bandwidth Justification (Reviewer dQEq). We listed 10 published works (ICML, NeurIPS, VLDB, SIGCOMM, ASPLOS) that evaluate training under sub-1Gbps settings.
- Applicability to TP / EP / DP (Reviewer W7By). We clarified how TAH-Quant naturally extends to tensor and expert parallelism, and discussed alignment challenges for data-parallel weight/gradient quantization.

We believe these additional experiments and clarifications significantly strengthen the paper and comprehensively respond to the core concerns raised by the reviewers.

Best regards,

The Authors of Submission 12668

---

### Meta-Review · Area_Chair_Rn9p · 2026-01-04

**Summary:**

This paper proposes TAH-QUANT, an activation quantization method for pipeline parallel training. The activation quantization involves 3 components 1) Tile-wise group quantization: partitions activation to small tiles and quantize each tile independently 2) Token-level adaptive bit allocation: entropy-based strategy that ranks of activations where top-p% of the tokens are quantized to INT4 and the rest to INT3, 3) Outlier suppression. The paper provides theoretical analysis of the method including convergence rate of $O(1/sqrt{T})$ and empirical results for fine-tuning GPT-2XL (1.5B parameters) and pretraining and fine-tuning of Qwen2.5-3B. The setup includes 8 instances each with a single GPU.

Summary of strengths by reviewers:
- Reviewer dQEq found the combination of ideas in the paper novel, the empirical evaluations clear and the convergence proof rigorous.
- Reviewer W7By found the proposed techniques novel and the theory good and the speedup of 1.33x convincing.
- Reviewer Nj5V found the problem practical, the method simple, and the theoretical analysis strong.
- Reviewer 7EhT found the theoretical convergence analysis a strength.

**Reviewer Concerns:**

Reviewer dQEq:
- **Computational cost**: Authors provided complexity analysis and runtime profiling (to be added to the paper).
- **Analysis of memory reduction vs compression cost vs batch size vs bandwidth**: The reviewer recommended including a micro-benchmark demonstrating the speed up. Authors provided micro-benchmarks with sweeps and speedups (to be added to the paper).
- **Bit allocation ablation**: The authors describe their chosen default configuration and provide ablations for other configurations (to be added to the paper).
- **Detailed description of method**: Authors provided more details in comments (to be added to the paper).
- **Small-scale experiments**: The reviewer noted that evaluations on Qwen2.5-3B (3B parameters) and GPT-2XL (1.5B parameters) are small-scale. The authors presented new experiments using LLaMA-8B for 8B tokens.
- **Reviewer response during rebuttal**: The reviewer responded before the authors provided their final results and highlighted that those results including experiments on larger models is important.

Reviewer W7By:
- **Outlier detection is heuristic**: Authors acknowledge that the threshold is chosen empirically and provide ablations to assess the robustness of the design (to be added to the paper).
- **3090 GPU is not a powerful machine**: The authors argue their target scenario is decentralized training where computation is contributed by geographically distributed participants. The AC is not convinced by this argument (details in reviewers scores).

Reviewer Nj5V:
- **Comparison with AQ-SGD is missing zero-shot evaluations**: The authors argue AQ-SGD cannot operate in training settings where a single SFT epoch is used. The AC is not convinced by this argument (details in reviewers scores).
- **Clarification on loss curves in Figure 2**: The authors provide justification why TAH-Quant achieves similar loss curves to AQ-SGD. The AC is not convinced by this argument (details in reviewers scores).
- **Additional quantization settings**: Authors provide results for other quantization configurations.

Reviewer 7EhT:
- **Small-scale experiments**: The reviewer notes that the experiments are limited to small models and short training durations. The authors additionally provide results for training  LLaMA-8B on the OpenWebMath dataset for 8B tokens.
- **Outlier detection is heuristic**: Authors acknowledge that the threshold is chosen empirically and provide ablations to assess the robustness of the design (to be added to the paper).
- **Naive quantization in backward pass**: The authors argue there is 2x more time available for communication-computation overlap during backward pass and so the quantization is less important. They also provide additional results with ultra-low precision in backward pass.
- **Breakdown of end-to-end training time**: Authors provide analysis and timings.
- **Comparison to other methods**: The reviewer suggested a comparison with a few prior works. However, authors note that they were not able to find the named papers and that the review may have been partially generated by AI. The AC also could not find the mentioned papers.

**Reviewer Scores:**

Three reviewers gave a score of 4 (marginally below the acceptance threshold) while one reviewer gave a score of 6 (marginally above the acceptance threshold).

However, the AC remains unconvinced by the authors' justification of the experimental scale, a concern raised by multiple reviewers from various perspectives. The experiments are notably constrained by:
- GPU Type: Limited to NVIDIA RTX 3090.
- Number of GPUs: Only 8 GPUs in total.
- Training Duration: A maximum of 8 billion tokens, suggesting fine-tuning or post-training.
- Model Size: 3B in the paper, with an 8B model provided during the rebuttal.

While the authors cited 10 papers in their rebuttal to support their low-bandwidth setup, most of these cited works utilized more powerful hardware (A100/V100/H100) and/or a significantly larger scale (e.g., 8-32 nodes, each with 8 GPUs). The authors argue their setup targets "decentralized training where computation is contributed by geographically distributed participants”. Such a setup can be justified with either significantly more participants, larger models, or longer training.

In terms of evaluation, the results are primarily comparisons of loss curves, with only one setup including downstream evaluation (Table 3). Consequently, the authors are required to include more comprehensive downstream evaluations for all architectures and setups explored in the work.

The AC recommends authors to improve their experiments, clarify the target setup and preferably follow a setup from comparable papers, and incorporate responses in the paper.

---

### Decision · Program_Chairs · 2026-01-26

Reject